

# 1   Assessing acetone for the GISS ModelE2.1 Earth system model

Alexandra Rivera[1], Kostas Tsigaridis[2,3], Gregory Faluvegi[2,3], Drew Shindell[4]
[1]Pratt School of Engineering, Duke University, Durham, NC, 27708, USA
[2]Center for Climate Systems Research, Columbia University, 2880 Broadway, New York, NY, 10025, USA
[3]NASA Goddard Institute for Space Studies, 2880 Broadway, New York, NY, 10025, USA
[4]Nicholas School of the Environment, Duke University, Durham, NC, 27708, USA
*Correspondence to*: Kostas Tsigaridis (kostas.tsigaridis@columbia.edu)
**Abstract.** Acetone is an abundant volatile organic compound in the atmosphere with important influence on ozone and oxidation
capacity. Direct sources include anthropogenic, terrestrial vegetation, oceanic, and biomass burning emissions. Acetone is also
produced chemically from other volatile organic compounds. Sinks include deposition onto the land and ocean surfaces, as well as
chemical loss. Acetone's lifetime is long enough to allow transport and reactions with other compounds remote from its sources.
The latest NASA Goddard Institute for Space Studies (GISS) Earth System Model, ModelE2.1, simulates a variety of Earth system
interactions. Previously, acetone had a very simplistic representation in the ModelE chemical scheme. This study assesses a more
sophisticated acetone scheme, in which acetone is a full 3-dimensional tracer, with explicit sources, sinks and atmospheric
transport. We evaluate the new global acetone budget in the context of past literature. Anthropogenic emissions, vegetation
emissions, biomass burning, and deposition representations agree well with previous studies. Chemistry and the ocean contribute
to both sources and sinks of acetone, with their net values agreeing with the literature, although their individual source and sink
terms appear to be overestimated for chemistry and underestimated for ocean fluxes. We find the production of acetone from
precursor hydrocarbon oxidation has strong leverage on the overall chemical source, indicating the importance of accurate molar
yields for this source. Spatial distributions reveal that ocean uptake of acetone is strongest in northern latitudes, while production
is mainly in mid-southern latitudes. The seasonality of acetone-related processes was also studied in conjunction with field
measurements around the world. These comparisons show promising agreement, but have shortcomings at urban locations, since
the model's resolution is too coarse to capture behavior in high-emission areas. Overall, our analysis of the acetone budget aids
the development of this tracer in the GISS ModelE2.1, a crucial step to understanding the role of acetone in the atmosphere.

## 25   1 Introduction

Acetone ($C_3H_6O$) is an abundant oxygenated volatile organic compound (VOC) that has important connections to ozone and the
atmosphere's self-cleansing oxidation capacity (Read et al., 2012). Acetone's dynamic presence in Earth's atmosphere can be
described through sources, sinks, and mechanisms of transport. Extensive literature has discussed the nature of these sources and
sinks, and some are more well-constrained than others.
Primary sources of acetone in the atmosphere include anthropogenic, terrestrial vegetation, and biomass burning emissions. Past
literature has found the fluxes of these sources to range between 1-2 Tg yr[-1], 30-45 Tg yr[-1], 2.5-4.5 Tg yr[-1], respectively (Beale et
al., 2013; Brewer et al., 2017; Elias et al., 2011; Fischer et al., 2012; Folberth et al., 2006; Jacob et al., 2002; Singh et al., 2000;
Wang et al., 2020). Chemical production from other VOCs with 3 or more carbon atoms, each with their own molar yields, is
another source of acetone in the atmosphere (Brewer et al., 2017; Fischbeck et al., 2017; Hu et al., 2013; Jacob et al., 2002; Singh
et al., 2000; Weimer et al., 2017).






Sinks of acetone include wet and dry deposition onto the land surface, as well as chemical loss. Wet deposition occurs within and
below clouds, due to the high solubility of acetone, which depends on its Henry's Law coefficient (Benkelberg et al., 1995). Dry
deposition occurs on the land surface. Chemical loss of acetone forms radicals, either through oxidation by OH or photolysis. Past
literature has estimated the acetone sinks to be 10-30% dry deposition, and 40-85% chemical loss (Arnold et al., 2005; Elias et al.,
2011; Fischer et al., 2012; Khan et al., 2015; Singh et al., 1994). The estimated fluxes are 10-16 Tg yr$^{-1}$ and 45-60 Tg yr$^{-1}$ for total
deposition and chemical loss, respectively (Arnold et al., 2005; Brewer et al., 2017; Dufour et al., 2016; Elias et al., 2011; Fischer
et al., 2012; Jacob et al., 2002; Khan et al., 2015; Marandino et al., 2005; Singh et al., 2000; Wang et al., 2020).

The ocean surface is a bidirectional flux that provides both a source and a sink for acetone. Ocean surface conditions such as wind
speed, sea surface temperature, and seawater concentration of acetone can influence the direction and magnitude of ocean-acetone
exchange (Wang et al., 2020). Previous literature estimates an oceanic source flux of 25−50 Tg yr$^{-1}$ and oceanic uptake flux of
35−60 Tg yr$^{-1}$. However, there is little consensus in the literature on whether the ocean serves as a net source or sink of acetone,
with some studies indicating a net oceanic source (Beale et al., 2013; Jacob et al., 2002; Wang et al., 2020), and other studies
indicating a net oceanic sink (Brewer et al., 2017; Elias et al., 2011; Fischer et al., 2012; Wang et al., 2020).

In addition to a global annual mean atmospheric budget, previous studies have reported the seasonality of acetone-related processes.
Past studies have compared monthly estimates of acetone mixing ratios to field measurements of European sites from Solberg et
al. (1996) (Arnold et al., 2005; Elias et al., 2011; Jacob et al., 2002). Comparisons with these European sites have emphasized the
seasonal variability of acetone emissions, as nearly all sites portray a summer maximum and winter minimum of acetone
abundance. Vegetation emissions from June to September, along with chemical sources, have an especially strong contribution to
this seasonality. The winter minimum of acetone is aided by an ocean sink at coastal sites (Jacob et al., 2002).

Other studies have described spatial distributions and seasonal dependence of ocean fluxes of acetone (Fischer et al., 2012; Wang
et al., 2020). A model by Fischer et al. (2012) proposed a net ocean sink of 2 Tg yr$^{-1}$, characterized ocean uptake of acetone as
strongest in northern latitudes year-round, and strongest in the high southern latitudes during the winter. An oceanic acetone source
was dominant in the tropical regions, with an exception off the Western coasts of Central America and Central Africa (Fischer et
al., 2012). A model by Wang et al. (2020) that varied surface seawater acetone concentration through a machine learning approach
also proposed a net ocean sink year-round. This net sink was strongest in December-February, and weakest in March-May.

The vertical distribution of acetone has been modelled between the seasons of May-October and November-April in the surface
and troposphere (Fischer et al., 2012). Acetone concentrations are generally higher in the lower altitudes due to proximity to surface
emissions. Surface-level acetone has been measured over a variety of terrestrial and oceanic sites around the world (de Gouw et
al., 2004; Dolgorouky et al., 2012; Galbally et al., 2007; Guérette et al., 2019; Hu et al., 2013; Huang et al., 2020; Langford et al.,
2010; Lewis et al., 2005; Li et al., 2019; Read et al., 2012; Schade & Goldstein, 2006; Singh et al., 2003; Solberg et al., 1996;
Warneke & de Gouw, 2001; Yoshino et al., 2012; Yuan et al., 2013), and in some cases, these measurements were taken over a
variety of months to provide a sense of seasonality (Dolgorouky et al., 2012; Hu et al., 2013; Read et al., 2012; Schade & Goldstein,
2006; Solberg et al., 1996). Additionally, vertical distributions of acetone have been measured through NASA's Atmospheric
Tomography Mission (ATom) campaigns (Thompson et al., 2022). The ATom-1, ATom-2, ATom-3, and ATom-4 campaigns took
place during July-August 2016, January-February 2017, September-October 2017, and April-May 2018, respectively. Each





campaign provided mixing ratios for a variety of VOCs in profiles from the marine boundary layer up to the upper
troposphere/lower stratosphere (Apel et al., 2021).

The NASA Goddard Institute for Space Studies (GISS) ModelE2.1 Earth System Model (Kelley et al., 2020) has the capability of
simulating a variety of Earth system interactions, is used both to interpret and predict past and future climate, and routinely
participates in the Climate Model Intercomparison Projects (CMIP) and Intergovernmental Panel for Climate Change (IPCC)
reports. Here we used this model and enhanced it by adding acetone as an independent chemical tracer (Kelley et al., 2020).
Previously, acetone had a very simplistic representation in the model's chemical scheme (Shindell et al., 2003), in which acetone's
spatial variation was parameterized based on the difference of the model's zonal mean distribution of isoprene and that tracer's
three-dimensional distribution. Acetone's lifetime is long enough to be transported remote from sources, but not long enough to
become uniformly mixed, and therefore its simulated distribution should benefit from a more realistic implementation. We
developed a greatly improved acetone tracer scheme by making prognostic calculations of the 3-dimensional distribution of acetone
as a function of time. We evaluated its atmospheric burden and lifetime as well as source/sink fluxes (anthropogenic emissions,
vegetation emissions, biomass burning, deposition, ocean, and chemistry) against other models and its concentration against field
measurements. This work aims to provide a holistic assessment of the abundance of acetone in the atmosphere.
**2 Methodology**
Our 'Baseline' simulation is a climatological mean with year 2000 conditions, chosen to be relatively modern without precluding
comparison with models in older literature. The 1996-2004 mean of prescribed emissions from Hoesly et al. (2018) were used,
along with the 1996-2005 mean sea surface temperature and sea ice cover as described in Kelley et al., (2020). An additional
simulation, 'Nudged_ATom', was conducted to compare more directly with ATom field measurements. This simulation employed
nudged winds (from MERRA2) (Gelaro et al., 2017) and ocean surface conditions and trace gas and aerosol emissions changing
with time during 2016-2018.
**2.1 Sources**
**2.1.1 Anthropogenic emissions**
Anthropogenic emissions were prescribed using the 1996-2004 averages of the Community Emissions Data System (CEDS)
emissions from Hoesly et al. (2018) as prepared for the GISS contributions to the Coupled Model Intercomparison Project, Phase
6 (CMIP6) (Kelley et al., 2020). These include sources from agriculture, the energy sector, the industrial sector,
residential/commercial/other, international shipping, solvents production and application, the transportation sector, and waste. In
line with past studies, we base acetone emissions on that of ketones. VOC23-ketones emissions from Hoesly were scaled down by
a ratio of acetone molecular weight to an average ketone molecular weight (58.08 g mol$^{-1}$/75.3 g mol$^{-1}$). Maintaining the resulting
spatial and temporal pattern of emissions, the magnitudes were then tuned to be close to that of Fischer et al., (2012), resulting in
a total of about 1 Tg yr$^{-1}$. This resulted in roughly 36.5% of CEDS VOC23-ketones used as acetone emissions. Lacking an accurate
way to obtain acetone aircraft emissions from the bulk VOCs available in the emission inventory, we have neglected that sector in
the simulations.



### 2.1.2 Terrestrial vegetation emissions

Emissions from land vegetation were derived from the Model Emissions of Gases and Aerosols from Nature (MEGAN), version 2.1 (Guenther et al., 2012), a new contribution to the ModelE. Emission response algorithms in the MEGAN2.1 model are derived from input leaf area indices, solar radiation, temperature, moisture, $CO_2$ concentrations, and plant functional types and composition of species (Guenther et al., 2012). The acetone vegetation emissions in the Baseline simulation in GISS ModelE2.1 are calculated to equal 36.1 Tg yr$^{-1}$.

### 2.1.3 Biomass burning emissions

Acetone emissions were prescribed from a 1996-2004 average of the NMVOC-C3H6 species from version 2.1 of the biomass burning dataset of van Marle et al. (2017), used by CMIP6. The acetone mass flux from biomass burning in the Baseline simulation was 1.59 Tg yr$^{-1}$.

Figure 1 shows the biomass burning emission rate chosen for this study, and how it lies within the range of substantial interannual variability. During the 20-year period shown, emissions averaged 1.463 Tg yr$^{-1}$, with a standard deviation of 0.402, and a spike in the earlier years of emissions over 2.75 Tg yr$^{-1}$ is also observed (Figure 1). On top of any differences across emission inventories, the years considered when reporting emissions may be the reason for variability between models (e.g. 2.40 – 2.80 Tg yr$^{-1}$ from the 2006 GFED-v2 emission inventory in Elias et al. (2011) and Fischer et al. (2012), compared to 3.22 Tg yr$^{-1}$ from 1997-2001 in Folberth et al. (2006)).

### 2.2 Sinks

#### 2.2.1 Deposition

Both dry and wet deposition of acetone were included in the model, although dry deposition was, on average, 91% of total deposition. The wet deposition scheme is given by Koch et al., (1999). Acetone and other species are transported within and below clouds, and soluble gases are deposited depending on the conditions of the grid box they are in and a Henry's Law Coefficient (Shindell et al., 2001). The Henry's Law Coefficient for acetone used in the GISS ModelE2.1 is 27 mol L$^{-1}$ atm$^{-1}$, with a Henry temperature dependence of acetone of 5300 J mol$^{-1}$ (Benkelberg et al., 1995; Zhou & Mopper, 1990). The dry deposition scheme uses resistance-in-series calculations, global seasonal vegetation data (Chin et al., 1996; Shindell et al., 2001; Wesely & Hicks, 1977), and a reactivity factor of $f_0$=0.1. This resulted in an acetone deposition rate in the Baseline simulation of 22.2 Tg yr$^{-1}$.

### 2.3 Chemistry

The GISS ModelE2.1 Baseline simulation estimates a net chemistry change of -20.6 Tg yr$^{-1}$. The components can be broken up into sources and sinks as follows.

#### 2.3.1 Chemical sources

The Baseline simulation estimates chemical production to be 33.3 Tg yr$^{-1}$. The acetone chemical scheme includes two production reactions:

$$Paraffin + OH \rightarrow 0.35 \ Acetone \qquad (1)$$

$$Terpenes + \{OH, O_3\} \rightarrow 0.12 \ Acetone \qquad (2)$$



In the first reaction, acetone is produced by paraffin, a proxy tracer for paraffinic (saturated) carbon, and OH (Eq. 1). The molar
yield of acetone from paraffin was found to be a strong leverage to the overall chemical source (see Section 3.5). A rate coefficient
of 8.1E-13 cm$^3$ molecule$^{-1}$ s$^{-1}$ was used (Shindell et al., 2003). Previous literature has suggested an acetone yield on a molecular
scale of 0.72 (Fischbeck et al., 2017; Jacob et al., 2002; Weimer et al., 2017). Initial tests using a yield of 0.72 resulted in an
overestimated chemistry source, leading us to re-evaluate this yield for the specific mixture of VOCs represented in the GISS
ModelE2.1. Estimated mole fractions of propane (11%), butane (22%) and pentane (21%) in anthropogenic emissions were
multiplied by each compound's acetone molar yield (0.73, 0.95, 0.63, respectively), determining that 42% of paraffin from
anthropogenic sources becomes acetone. Estimated mole fractions of propane (9%) and higher alkanes (23%) in biomass burning
emissions were multiplied by each compound's acetone molar yield, determining that 25% of paraffin from biomass burning
sources becomes acetone. The molar yields used in these calculations were derived with suggestions from the literature (Fischbeck
et al., 2017; Jacob et al., 2002; Weimer et al., 2017). An average of the 42% anthropogenic paraffin and 25% biomass burning
paraffin was used to conclude that approximately 35% of paraffin from emissions becomes acetone, leading to the molar yield of
0.35 in Eq. (1).
Additionally, reactions between terpenes and {OH, O$_3$} were implemented with an acetone yield of 0.12 (Hu et al., 2013; Jacob et
al., 2002) (Eq. 2). The rates for these reactions are 2.51E-11*exp(444/T) cm$^3$ molecule$^{-1}$ s$^{-1}$ for the OH reaction and 1.40E-14*exp(-
732/T) cm$^3$ molecule$^{-1}$ s$^{-1}$ for the O$_3$ reaction, and these coefficients are enhanced from the standard α-pinene one to consider the
reactivity variability across mono- and higher terpenes (Tsigaridis and Kanakidou, 2003).
**2.3.2 Chemical sinks**
The chemical sink of acetone in the Baseline simulation is estimated to be 53.8 Tg yr$^{-1}$. The sinks of acetone include oxidation by
OH and Cl radicals, and photolysis:
$Acetone + OH \rightarrow H_2O + CH_3C(O)CH_2$ (assumed to decompose to HCHO) (3)
$Acetone + Cl \rightarrow HCl + CH_3C(O)CH_2$ (assumed to decompose to HCHO) (4)
$Acetone + hv \rightarrow CH_3CO + CH_3$ (5)
$Acetone + hv \rightarrow CH_3 + CH_3 + CO$ (6)
The first and second acetone destruction reactions above have rates of 1.33E-13 + 3.82E-11*exp(-2000/T) cm$^3$ molecule$^{-1}$ s$^{-1}$ and
7.70E-11*exp(-1000/T) cm$^3$ molecule$^{-1}$ s$^{-1}$, respectively (Sander et al., 2011) (Eq. 3, 4). Previously, acetone photolysis (which only
affected production of radicals and not acetone itself) did not utilize the model's photolysis scheme but was parameterized solely
as a function of orbital geometry and atmospheric pressure. In the model updates, photolysis now consists of two separate reactions,
where acetone forms either CH$_3$CO + CH$_3$ radicals or two CH$_3$ radicals and CO (Eq. 5, 6). Reaction 5 is pressure-dependent, while
reaction 6 is temperature-dependent.
**2.4 Ocean**
Bidirectional fluxes of acetone are calculated over ocean based on the "two-phase" model of molecular gas exchange at the air-sea
interface of Liss & Slater (1974), as it is described in Johnson (2010). The fluxes are a function of simulated surface temperature
and near-surface wind speed but independent of salinity. Henry's Law constants and temperature dependence of solubility for
acetone are from Sander (1999). The atmospheric source from ocean water and sink from the atmosphere are calculated assuming
a constant concentration of acetone in water (of 15 nM), the lower boundary layer atmospheric concentration, and the total transfer



velocity (a combination of water-side and air-side transfer velocities). The GISS ModelE2.1 Baseline simulation calculates the
ocean to be a net source of acetone, producing 3.94 Tg yr$^{-1}$.

**2.5 Sensitivity studies**

Sensitivity studies were conducted to determine the influence of key parameters on the acetone budget and its global distribution
(Table 1). Specifically, we were interested in seeing how much leverage a given parameter afforded the model by way of an
artificial perturbation. Sensitivity studies for chemistry modify the sources of acetone. The Chem_Cl0 and Chem_Terp0
simulations provide no formation of acetone from chlorine or terpenes, respectively (Table 1). The importance of paraffin is
explored by halving its yield of acetone to 17.5% in the Chem_Par0.5 simulation, and by doubling its yield of acetone to 70% in
the Chem_Par2.0 simulation (Table 1). As vegetation was the most prominent source, the Veg_0.7 simulation observes its
reduction by decreasing the MEGAN production of acetone by 30% (Table 1). The Ocn_2.0 simulation aims to explore the impact
of ocean acetone concentration by doubling it from 15 nM to 30 nM globally (Table 1). The Dep_$f_0$0 simulation tested dropping
the reactivity factor for dry deposition from 0.1 to zero. Finally, given the high interannual variability of biomass burning emissions,
the BB_2.0 simulation explores the impact of doubling those emissions (Table 1).

**3 Results and model evaluation**

**3.1 Global acetone budget and burden**

A global acetone budget table was compiled from past global modeling studies to place our estimates in context (Table 2) (Arnold
et al., 2005; Beale et al., 2013; Brewer et al., 2017; Dufour et al., 2016; Elias et al., 2011; Fischer et al., 2012; Folberth et al., 2006;
Guenther et al., 2012; Jacob et al., 2002; Khan et al., 2015; Marandino et al., 2005; Singh et al., 2000; Singh et al., 2004; Wang et
al., 2020). The values of the individual fluxes in our model (global deposition, biomass burning, anthropogenic emissions,
vegetation emissions, ocean net/source/sink, and chemistry net/source/sink) were mentioned previously.

Atmospheric burden describes the total amount of acetone that is in the atmosphere. The GISS ModelE2.1 Baseline simulation
estimates the burden to be 2.93 Tg yr$^{-1}$. Additionally, chemical lifetime and atmospheric lifetime can be derived from burden. The
chemical lifetime of acetone is calculated as the burden divided by the chemical sink, whereas total lifetime is the burden divided
by all sinks. The chemical and total atmospheric lifetimes for the Baseline simulation are calculated to be 19.9 and 12.3 days,
respectively. These values are also placed in the context of previous literature in Table 1.

The GISS ModelE2.1 Baseline acetone budget is further compared to previous model studies in Figure 2. The calculated fluxes in
our Baseline simulation that are less than one standard deviation away from the literature mean include anthropogenic and
vegetation emissions, net ocean, net chemistry, chemical production, and chemical destruction (Figure S1). Biomass burning in
GISS ModelE2.1 appears as an outlier when compared against 9 previous model studies but can be attributed to the high interannual
variability with emissions (as discussed in Section 2.1.3). The value of acetone deposition is on the high (more negative) end in
GISS ModelE2.1 relative to 11 previous studies. This might be partially attributed to differences in deposition parametrization
across models, as explored by our sensitivity study on dry deposition presented in section 3.5.2. The values for oceanic acetone
sources and losses are smaller (in absolute values) than the mean from 7 previous model studies. Nevertheless, the net ocean flux
matches the literature well. Lastly, the total atmospheric burden and lifetime calculated by GISS ModelE2.1 are lower than the
previous papers, an expected consequence of the higher removal by deposition. The chemical lifetime is also calculated to be at



the low end of published literature. As the burden is a function of many different atmospheric parameters, however, it was not the
goal to corroborate our estimates with the literature as much as it was for each of the fluxes.

## 3.2 Spatial distribution of acetone

The global distribution of acetone at the surface is given in Figure 3. It is evident that acetone mixing ratios are largest over the
continents, where anthropogenic, vegetation, and other terrestrial sources are located. Over the ocean, acetone mixing ratios are
highest downwind of central America and central Africa. A comparison of the GISS ModelE2.1 results against twenty-six prior
field measurements shows an overall great agreement, with a root mean squared error of 0.3494 and an $R^2$ value of 0.8306. To put
these results into the context of model evaluation, a similar comparison to field measurements was done for the model's previous
acetone scheme. The prior parameterization was designed as a rough representation of acetone oxidized from isoprene in the upper
troposphere, without regard for realism near the surface, and this is evident from the comparison with surface observations: a root
mean squared error and $R^2$ value of 1.3620 and 0.0413, respectively. The improvement of the new acetone tracer model in the
GISS ModelE2.1 is evident from these statistics.

A breakdown of the acetone bidirectional fluxes indicates that its chemical production is concentrated over the continents, while
chemical destruction is primarily over the oceans (Figure 4). Hotspots of production over the continents include the Southern and
Eastern United States and central South America, East and Northern Asia, and Central Africa. Chemical sinks over the oceans are
stronger in the tropics than in the high southern or northern latitudes. Annually, there is a net negative flux of about -20.46 Tg yr-
$^1$ (Figure 4). Observing the chemical flux over all four seasons, the net loss appears unaffected while the net source changes more
significantly, following the seasonality of precursor compounds like isoprene and terpenes (Figure 5). Chemical production is
strongest in the months of June/July/August, primarily in the US and Northern Asia. Production is weakest in the months of
December/January/February, losing almost all production in the US and Northern Asia entirely. Still, a net negative flux is present
for all four seasons (Figure 5).

The ocean acetone sources and sinks are unevenly distributed across latitudes. Oceanic uptake of acetone is mostly concentrated
in the northern rather than the southern oceans, while the ocean acetone source is strongest in the tropics and decreases at higher
latitudes of both hemispheres (Figure 6). Combining these two unidirectional fluxes results in the ocean serving as a sink in the
northern high latitudes, a source in the tropical latitudes, and near neutral at the high southern latitudes (Figure 7). This finding
corroborates very well with findings from Fischer et al. (2012) and Wang et al. (2020). Oceanic bidirectional acetone fluxes present
trends over the four seasons (Figure S2). Overall, every season has a positive global mean net flux. However, production becomes
strongest in the months of December through May, and weakest in the months of June through November. Off the coast of western
South America, the ocean appears to be a net sink of acetone, even though this latitude band is generally a source of acetone. This
is especially evident in the months of June/July/August and September/October/November. As the model simulates this location
to have high levels of acetone at the surface (Figure 3), we believe the acetone in the air is driving the ocean to be a sink there
(Figure S2).

## 3.3 Vertical distribution of acetone

The vertical distribution of acetone varies by latitude, with near-surface air mixing ratios being higher in the tropics and in the
northern midlatitudes. Acetone levels in the atmosphere decrease with height, a direct result of sinks dominating the sources (Figure
8). Prior to the implementation of an acetone tracer in the GISS ModelE2.1, when acetone was derived from the zonal mean of





isoprene, the vertical distribution looked very different (Figure S3). Acetone was only concentrated around the tropics and did not
extend nearly as high into the atmosphere. The complexity of Figure 8 supports the new acetone tracer scheme as a significant
improvement to the GISS ModelE.

Another modelled vertical distribution of acetone, including a differentiation between two long seasons, is explored in Figure 9.
In general, it was found that acetone mixing ratios are higher in the months of May-October than in November-April, and that this
relationship is stronger in the lower atmosphere (0-2 km) than the upper atmosphere (6-10 km). This finding corroborated well
with a similar analysis done by Fischer et al. (2012).

Additionally, the GISS ModelE2.1 was compared to four ATom campaigns (Thompson et al., 2022) of acetone field measurements
in the atmosphere (Apel et al., 2021). For this comparison, we averaged the flight data to the model grid, and then compared the
resulting mean against the monthly mean fields of the model output. Contrary to other chemical species measured during ATom
that vary significantly in space and time, acetone has a rather long lifetime, and the data are collected for the most part very far
from its sources. Combining that with the fact that prescribed emissions in the model vary by month, not by day or even hour in
GISS ModelE2.1, makes such a comparison appropriate. Meteorology though can affect long-range transport significantly, so for
that reason we performed a nudged simulation (called Nudged_ATom) towards the MERRA-2 reanalysis (Gelaro et al., 2017), to
capture such an effect more accurately. We also used emissions and greenhouse gas concentrations from the years of the ATom
campaigns and varying with year, rather than the climatological means used in the Baseline simulation. Both the Nudged_ATom
and Baseline simulations are plotted in the ATom comparisons presented here (Figure 10). Although there are some differences at
times, for example in the tropical Atlantic Ocean, for the most part the two simulations are indistinguishable, indicating that our
conclusions comparing climatological simulations to ATom should be robust. (Figure 10, Figures S4-S6). The GISS ModelE2.1
was found to match measurements particularly well in the winter and fall seasons (ATom-2 and ATom-3, respectively). The model
underestimated measurements in the mid-northern latitudes in the spring and summer seasons (ATom-4 and ATom-1,
respectively), indicating that perhaps the model is not capturing a spring/summer source of acetone in the North. Generally,
however, the model matches remote atmosphere measurements remarkably well (Figure 10, Figures S4-S6).
**3.4 Seasonality of acetone**
Most European sites presented in Figure 3 have monthly-resolved measurements that can be used to analyze the seasonal behavior
of acetone in the model (Figure 11, Figure S7) (Solberg et al., 1996). These sites differ with respect to their geographic locations
and their proximity to anthropogenic sources. Zeppelin, Birkenes, Rucava, and Mace Head are all coastal sites, while Waldhof,
Kosetice, Donon, Ispra, and Montelibretti are inland sites. Regarding anthropogenic sources, Zeppelin is the most remote location
and Birkenes and Rucava each have small sources. Mace Head is a site affected by the marine boundary layer, and Waldhof,
Kosetice and Donon are sites with small local anthropogenic sources that are generally located in higher emission regions.
Montelibretti and particularly Ispra are subject to the highest anthropogenic sources. The measurements taken at Ispra show an
opposite seasonality than what is expected, and previous studies have considered this anomalous (Jacob et al., 2002).

The GISS ModelE2.1 matches the seasonality of the measurements well, especially in Zeppelin, Mace Head, Waldhof, Kosetice,
and Donon. The root mean squared error (RMSE) between the Baseline model and measurements at these five sites are 0.1969,
0.0914, 0.3907, 0.3430 and 0.3160, respectively. The model overestimates the measurements in Birkenes and Rucava (RMSE ≅
0.87 for both), even though these two sites have low anthropogenic sources. This overestimation has been attributed to the





vegetation source, which has a distinct seasonality and is much stronger than any other source there. Interestingly, in Montelibretti,
the model's overestimation of vegetation, yet underestimation of local emissions, results in a decent estimation of the sources there
(RMSE = 0.5454) (Figure 11).

As mentioned previously, an analysis of the distribution of the regional sources and sinks at the nine European sites shows that,
except for Zeppelin and Mace Head, all studied European sites have vegetation as the dominant source that strongly contributes to
the simulated seasonality of concentrations (Figure 12). Vegetation sources peak in the summer months and are lower in the winter.
Deposition is a major sink of acetone that is comparable in magnitude with the vegetation source. Ocean uptake of acetone follows
a weak seasonal cycle, being stronger in the summer months. The other fluxes (anthropogenic emissions, biomass burning and
ocean production) do not exhibit much seasonality at these locations (Figure 12).

We also compared the GISS ModelE2.1's surface acetone at observation sites with less temporal coverage (Figure 13) (de Gouw
et al., 2004; Dolgorouky et al., 2012; Galbally et al., 2007; Guérette et al., 2019; Hu et al., 2013; Huang et al., 2020; Langford et
al., 2010; Legrand et al., 2012; Li et al., 2019; Read et al., 2012; Schade & Goldstein, 2006). In general, the GISS ModelE2.1
matches the field measurements well. This is especially true for the non-summer seasons in Rosemount and Berkeley, and the
summer peaks in Utrecht and Mainz. The model seems to be overestimating acetone around Australia, as shown by comparisons
with Cape Grim and Wollongong, while underestimating emissions in large cities like Shenzhen, Beijing, London, and Paris.
**3.5 Sensitivity studies**
The sensitivity simulations presented here have been described in section 2.5 and in Table 1. We grouped them in two categories:
those directly related with chemical sources and sinks, and those related with terrestrial and oceanic acetone fluxes. Overall, the
sensitivity studies that presented large changes to total atmospheric burden included Chem_Terp0, Chem_Par0.5, Chem_Par2.0,
Veg_0.7, Ocn_2.0, and Dep_f$_0$0 (all but Chem_Cl0 and BB_2.0) (Figures S8-S13).
**3.5.1 Chemistry**
Chemistry sensitivity tests that modified the sources of acetone were analyzed with respect to the budget and global distribution
of acetone. In the Chem_Cl0 simulation, where no acetone oxidation by the chlorine radical occurs, the overall global acetone
budget does not change. However, in some places like Rucava, Ispra, Montelibretti, and Shenzhen, the shape of the acetone
concentration profile over the year changes slightly (Figure 14, Figure S14). The Chem_Terp0 simulation that removes the
production of acetone from terpenes decreases the summer peak of acetone by as much as 35.5% in Birkenes, 25.5% in Mainz,
and 25.3% in Berkeley (Figure 14, Figure S14). Other sites like Montelibretti, Ispra and Paris have their summer peak decreased
by 22.6%, 22.2%, and 19.0%, respectively (Figure 14, Figure S14). Coastal and remote areas like Zeppelin, Mace Head and
Dumont d'Urville are not impacted by the removal of terpenes (Figure 14, Figure S14). There seems to be some nonlinearities
with the relationship between acetone abundance and its yield from paraffin, as the results from the Chem_Par2.0 and Chem_Par0.5
simulation reveal that doubling the yield has a stronger impact than halving it. For instance, in Montelibretti, doubling the yield
from paraffin increases the summer peak by 35.7%, while halving the yield decreases the summer peak by only 8.3% (Figure 14,
Figure S14). A similar relationship is observed at other sites: Ispra (19.1% increase with double paraffin, 2.5% decrease with half
paraffin) and Berkeley (12.7% increase with double paraffin, 2.5% decrease with half paraffin) (Figure 14, Figure S14). Overall,
we explored chemistry sensitivities that would tend to push acetone in both directions. The Baseline simulation falls between our
tests, which we have identified as important uncertainties.






The spatial distribution differences between the chemistry sensitivity studies and the Baseline simulation show some interesting
patterns (Figure 15). Removing the production of acetone from terpenes oxidation decreased acetone over the continents, and
especially over tropical and boreal forests which are where terpenes are emitted. This change induced a feedback where acetone
concentration increased slightly over the oceans (Figure 15, top left). Halving production of acetone from paraffin oxidation only
decreased acetone concentrations over the continents (Figure 15, top right), while doubling it increased acetone concentrations
over the continents but reduced it marginally downwind (Figure 15, bottom). Feedback resulting from this change was that acetone
destruction increased over the tropics.
**3.5.2 Terrestrial and oceanic fluxes**
Terrestrial and oceanic fluxes sensitivities were analyzed at the same sites. The vegetation flux sensitivity, Veg_0.7, reduced
acetone production from MEGAN by 30%. This change decreased the summer peak of acetone down at nearly every location
studied, but most notably by 32.6% in Birkenes, 22.9% in Rucava, and 22.2% in Rosemount (Figure 16, Figure S15).

In the oceanic flux sensitivity simulation, Ocn_2.0, the concentration of acetone in the water was doubled from 15 nM to 30 nM.
The results of this simulation varied with geographic location. For instance, in Birkenes, doubling ocean concentration reduced
overall acetone by 13.9%, while in Montelibretti, it was increased by 16.1% (Figure 16). Even though Birkenes is more of a
coastal city than Montelibretti, this result may simply be a temperature effect; Birkenes is at 58°N, while Montelibretti is at
42°N, and a warmer ocean may produce more acetone. Overall, in most places, the doubling ocean acetone concentration did not
change much atmospheric acetone throughout the year.

Another broader finding from the ocean sensitivity study is that doubling the ocean acetone concentration impacted oceanic
emissions of acetone more than the oceanic uptake of acetone. Specifically, in this sensitivity study the emissions doubled, while
the uptake only increased by 40%. This difference may be attributed to the fact that a higher ocean concentration will generally
cause less resistance in the emission direction, but more resistance in the uptake direction. The differences in oceanic acetone
emissions and uptakes in this sensitivity study also resulted in increased chemical destruction, and an overall higher burden of
acetone in the atmosphere (Figure S12).

In the dry deposition sensitivity simulation, the reactivity factor, $f_0$, was reduced from 0.1 to 0. As a result, the amount of acetone
removed by deposition decreased, and the atmospheric acetone concentration increased. The strongest increases were found to be
in Ispra (38.4% increase), Kosetice (37.9% increase), Paris (37.9% increase), Beijing (37.3% increase), Donon (36.6% increase),
Mainz (33.4% increase), Montelibretti (30.5% increase), Rosemount (28.9% increase), Berkeley (28.7% increase), and Waldhof
(28.7% increase) (Figure 16, Figure S14). The final terrestrial fluxes sensitivity study, BB_2.0, doubled biomass burning emissions.
This sensitivity did not significantly change acetone mixing ratios in any of the locations studied, except an increased summer
spike (12.7% increase) in Birkenes (Figure 16). Most of the locations studied were far from biomass burning sites to begin with,
however, so an analysis of this sensitivity study over biomass burning hotspots is needed.

The acetone concentration anomalies around the world between the terrestrial and oceanic fluxes sensitivity studies and the
Baseline simulation are presented in Figure 17. Decreasing acetone production from MEGAN vegetation by 30% resulted in a
decrease of acetone mixing ratios over the tropical and boreal forests, where this source is most prominent (Figure 17, top left).





Doubling ocean acetone concentrations increased production of acetone from the oceans globally. This increase was stronger in the tropics, due to the higher sea surface temperatures (Figure 17, top right). Reducing the reactivity factor for dry deposition decreased the amount of acetone removed by deposition over the continents (Figure 17, bottom left), in particular where acetone concentration is elevated (Figure 3). Finally, doubling biomass burning emissions did not change acetone mixing ratios much, other than over biomass burning hotspots like central South America, central Africa, Southeast Asia, and Siberia (Figure 17, bottom right).

### 3.5.3 ATom comparisons

The ATom comparisons were replicated with the sensitivity simulations (Figure 18, Figures S16-S18). Doubling the paraffin yield and doubling the ocean acetone concentration seemed to have the most noticeable impacts on the vertical profiles. As seen in the summer season (ATom-1), doubling the paraffin yield more closely matches the measurements in the Northern hemisphere remote atmosphere (Figure 18). In the fall season (ATom-3), however, doubling the paraffin yield tends to overshoot most of the measurements (Figure S17). These results reveal that the model may be missing a paraffin source that is active during the summer season, and a paraffin sink that is active in the fall season. Additionally, the ocean sensitivity tends to shift the vertical profile right (overshooting measurements) in nearly all locations of the winter season (ATom-2), and in the Southern hemisphere of the other seasons. While the ocean flux may be small, these ATom comparisons reveal that they especially matter in the southern latitudes. These are the same latitudes where the ocean appears to be in equilibrium (neither a strong source nor sink) (Figure 7).

### 4 Conclusion

The development of acetone's representation in the NASA GISS ModelE2.1 from its previous simplistic parameterization of instantaneous isoprene to a full tracer experiencing transport, chemistry, emissions, and deposition of its own, marks a significant improvement to the model's chemical scheme. Calculations of the 3-dimensional distribution of acetone as a function of time, as well as evaluations of its atmospheric burden and source/sink fluxes demonstrate the complexity of acetone's spatiotemporal distribution in the atmosphere. An extensive analysis was conducted to assess the simulated global acetone budget in the context of past modeling studies. Further comparisons were made against field measurements on a variety of spatial and temporal scales, which indicated that the model agrees well with surface field measurements and vertical profiles in the remote atmosphere. The chemical formation of acetone from precursor compounds such as paraffin was found to be an uncertain yet impactful factor. Vegetation fluxes as calculated by MEGAN were identified as the dominant acetone source which dictates its seasonality. Additionally, the acetone concentration in seawater was found to affect oceanic sources more than oceanic sinks.

The work presented here demonstrates the usefulness of the approach to evaluate a chemical species in the model, and can be used for similar evaluations of other important gaseous and aerosol species. Any feedback between acetone and the rest of the chemistry, and particularly ozone, have not been assessed here, and should be the goal of a future study. Additionally, the current ocean-acetone interaction uses a constant concentration of acetone in the ocean. It will be helpful to test a more realistic, non-uniform ocean acetone concentration, when this becomes available. Finally, other atmospheric conditions such as surface wind speed may be considered further when modifying the ocean scheme.





**Code Availability**

The GISS ModelE code is publicly available at https://simplex.giss.nasa.gov/snapshots/. The most recent public version is E.2.1.2; the version of the code used here is already committed in the non-public-facing repository and will be released in the future following the regular release cycle of ModelE, under version E3.1. The code used for the simulations described here, including the configuration scripts ("rundecks") used to perform the simulations, are available together with the model output and scripts used to generate plots, here: https://doi.org/10.5281/zenodo.7954593.

**Data Availability**

We have made available the simulated three-dimensional distributions of acetone from each simulation described in the paper (Baseline, sensitivity simulations in Table 1, and Nudged_ATom). These are found in zip files, grouped by simulation, together with the model code and scripts used to generate plots, here: https://doi.org/10.5281/zenodo.7954593. Each zip file contains a series of netCDF format files with filenames {month}_5yrAvg_Acetone_{simulation}.nc, where each file is a climatological average over 5 years of repeated forcing conditions.

The exception is the transient-forcing simulation "Nudged_ATom", which contains single-month averages of acetone from JUL 2016 through MAY 2018, to cover the ATom observational period. The file names for that simulation are of the form: {month}_{year}_Acetone_Nudged_ATom.nc. Acetone is in ppbv units and given on the model's native grid and vertical levels. These are hybrid sigma levels, but nominal pressure middles and edges are given in the plm and ple variables, respectively, and the grid box surface areas are also provided.

**Author Contribution**

KT conceived the study and guided the model development which was done by GF. All simulations presented here were performed by GF. DS advised during the whole development process. AR did the literature search and all comparisons against other modeling studies. With the exception of the ATom analysis and plots which were done by KT, and comparisons against field measurements and the rest of the plots were done by AR. AR drafted the first version of the manuscript, and all authors contributed to it. GF prepared all model outputs for dissemination.

**Competing Interests**

The authors declare that they have no conflict of interest.

**Acknowledgements**

Climate modeling at GISS is supported by the NASA Modeling, Analysis and Prediction program. AR acknowledges support from North Carolina Space Grant and the NASA Office of STEM Engagement. Resources supporting this work were provided by the NASA High-End Computing (HEC) Program through the NASA Center for Climate Simulation (NCCS) at Goddard Space Flight Center.



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

**Tables and Figures**
**Table 1.** Sensitivity studies conducted to observe the leverage a specific parameter afforded the model. Simulation names, as well
as the parameter they target and a description, are included.

| GISS ModelE2.1 Sensitivity Simulation | Sensitivity Parameter | Description |
|---|---|---|
| Chem_Cl0 | Chemistry Source | Acetone + Chlorine reaction rate = 0 |
| Chem_Terp0 | Chemistry Source | No reaction for production of acetone from terpenes |
| Chem_Par0.5 | Chemistry Source | Half the yield of acetone from paraffin (17.5%) |
| Chem_Par2.0 | Chemistry Source | Double the yield of acetone from paraffin (70%) |
| Veg_0.7 | Vegetation | 0.7 factor of acetone from MEGAN |
| Ocn_2.0 | Ocean | Ocean acetone concentration from 15nM to 30nM |
| Dep_$f_0$0 | Dry Deposition | $f_0$ changed from 0.1 to 0 |
| BB_2.0 | Biomass Burning | Double biomass burning emissions |


**Table 2.** Global acetone budget table comparing burden, flux and lifetime estimates of acetone from the Baseline model to thirteen
previous studies.

| | Wang et al. [2020] [a] | Wang et al. [2020] [b] | Brewer et al. [2017] | Fischer et al. [2012] | Elias et al. [2011] | Jacob et al. [2002] | Other Estimates [2000-2016] [e] |
|---|---|---|---|---|---|---|---|




| Burden (Tg) | 3.5 | 3.80 | 5.57 | 5.60 | 7.20 | 3.80 | 3.50 – 4.20 |
|---|---|---|---|---|---|---|---|
| Global Deposition (Tg yr$^{-1}$) | -25.2 | -12.4 | -12.4 | -12.0 | -19.0 | -9.0 | -26.0 – -6.0 |
| Biomass Burning (Tg yr$^{-1}$) | 4.0 | 2.40 | 2.60 | 2.80 | 2.40 | 4.50 | 3.22 – 9.0 |
| Anthro Emissions (Tg yr$^{-1}$) | 0.50 | 3.40 | 3.60 | 0.73 | 1.60 | 1.10 | 1.02 – 2.0 |
| Vegetation Emissions (Tg yr$^{-1}$) | 39.8 | 32.2 | 37.1 | 32.0 | 76.0 | 35.0 | 15 – 56 |
| Net Ocean (Tg yr$^{-1}$) | -8.10 | 1.30 | -7.50 | -2.0 | -8.0 | 13.0 | 4.0 |
| Ocean Source (Tg yr$^{-1}$) | 33.4 | 45.7 | 51.8 | 80.0 | 20.0 | 27.0 | 20.0 |
| Ocean Sink (Tg yr$^{-1}$) | -41.5 | -44.4 | -59.2 | -82.0 | -28.0 | -14.0 | -62.0 |
| Net Chemistry (Tg yr$^{-1}$) | -11.1 | -26.1 | -22.5 | -21.0 | -53.0 | -45.0 | -33.0 – -5.50 |
| Chem Source (Tg yr$^{-1}$) | 38.5 | 26.1 | 24.1 | 31.0 | 27.0 | 28.0 | 15.5 – 55.6 |
| Chem Sink (Tg yr$^{-1}$) | -49.6 | -52.2 | -46.6 | -52.0 | -80.0 | -73.0 | -61.1 – -33.4 |
| Chemical Lifetime (days) [c] | 25.8 | 26.6 | 43.6 | 39.3 | 32.9 | 19.0 | 20.9 – 35.6 |
| Lifetime (days) [d] | 11.0 | 12.7 | 17.2 | 14.0 | 21.0 | 14.5 | 12.8 – 35 |

[a] CAM-Chem Model (Wang et al., 2020)
[b] GEOS-Chem Model (Wang et al., 2020)
[c] Chemical Lifetime = Burden/Chemical Sink
[d] Total Atmospheric Lifetime = Burden/Total Sink
[e] *Singh et al.* [2000, 2004], *Arnold et al.* [2005], *Folberth et al.* [2006], *Marandino et al.* [2006], *Guenther et al.* [2012], *Beale et al.* [2013], *Khan et al.* [2015], *Dufour et al.* [2016].



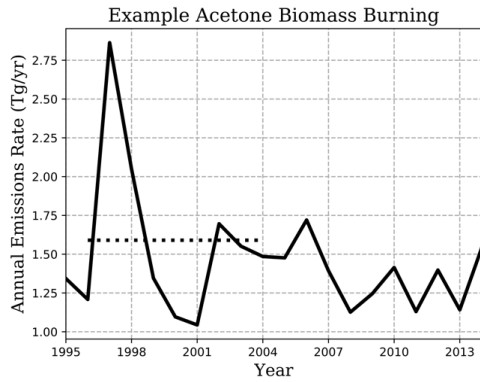


**Figure 1.** Illustration of interannual variability of NMVOC-C3H6 biomass burning emissions of van Marle et al., 2017 (solid line),
used as acetone emissions in our simulation. Climatological-emissions simulations use the 1996-2004 mean (dotted line), though
emissions vary with month.




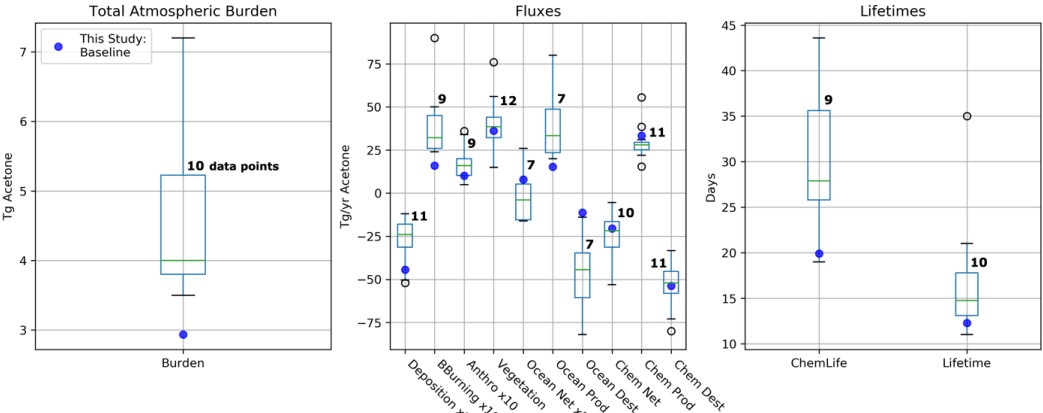


**Figure 2.** Total atmospheric burden, fluxes, and lifetimes of acetone from the literature (shown in boxes and whiskers with outliers
as open circles), and values from GISS ModelE2.1 (shown in solid circles). The number of models used to create each box and
whisker plot are labelled. Note that deposition and ocean net fluxes were multiplied by 2 and biomass burning and anthropogenic
emissions were multiplied by 10 for a better visualization of the distribution.

610

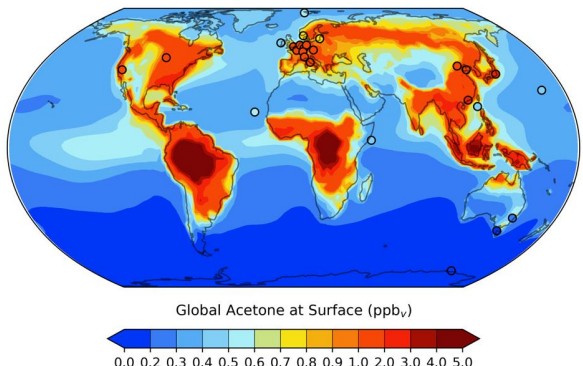

611

**Figure 3.** GISS ModelE2.1 spatial distribution of annual mean acetone at surface for the Baseline simulation. Filled circles
represent data from twenty-six field measurements (de Gouw et al., 2004; Dolgorouky et al., 2012; Galbally et al., 2007; Guérette
et al., 2019; Hu et al., 2013; Huang et al., 2020; Langford et al., 2010; Lewis et al., 2005; Li et al., 2019; Read et al., 2012; Schade
& Goldstein, 2006; Singh et al., 2003; Solberg et al., 1996; Warneke & de Gouw, 2001; Yoshino et al., 2012; Yuan et al., 2013).
The root mean squared error and the $R^2$ value between the Baseline acetone estimations and the field measurements are 0.3494
and 0.8306, respectively. A nonlinear colorbar is used to better differentiate the details in the map.





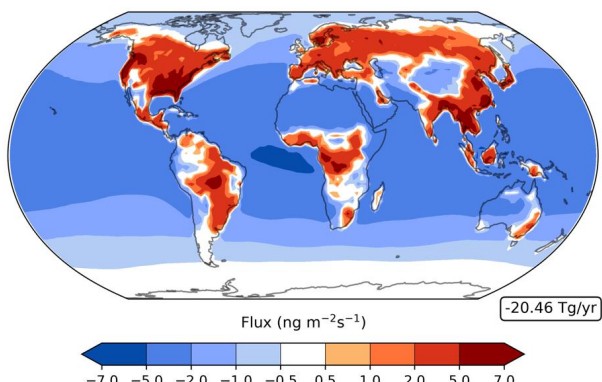

**Figure 4.** Annual average of acetone net chemistry fluxes (column-integrated) in the Baseline simulation, with red indicating a net source and blue indicating a net sink. A nonlinear colorbar is used to better differentiate the details in the map. The weighted global mean of the net chemistry flux is shown in a box on the lower right.

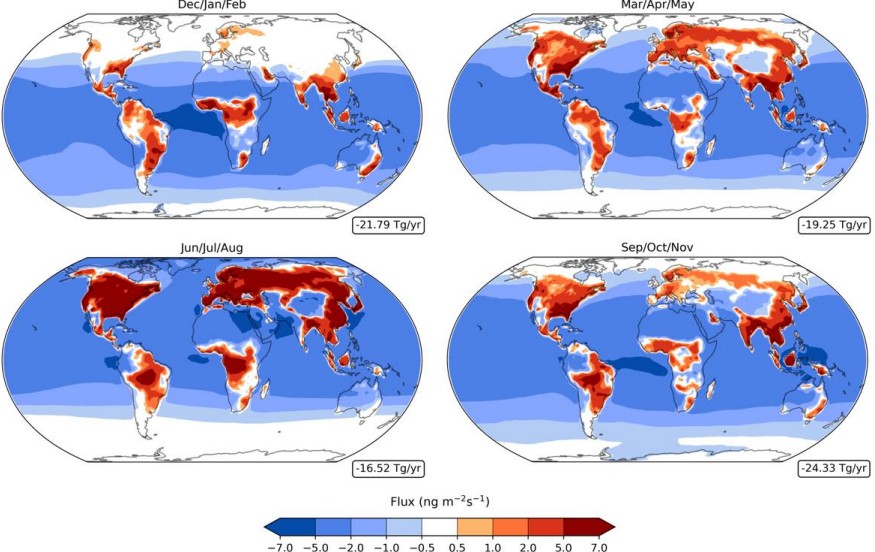

**Figure 5.** Acetone net chemistry fluxes (column-integrated) in the Baseline simulation for December-February (top left), March-May (top right), June-August (bottom left), and September-November (bottom right), with red indicating a net source and blue indicating a net sink. Nonlinear colorbars are used to better differentiate the details in the map. The weighted global means of the net chemistry fluxes are shown in boxes on the lower right.





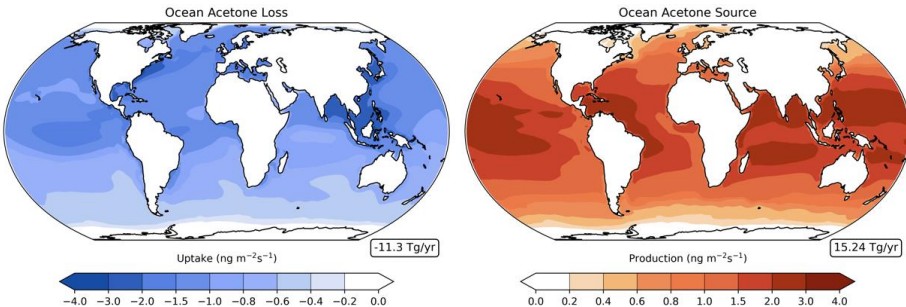

**Figure 6.** Annual average of the acetone ocean loss (left) and ocean source (right) in the Baseline simulation. Nonlinear colorbars are used to better differentiate the details in the map. The corresponding weighted global means of the ocean fluxes are shown in boxes on the lower right.

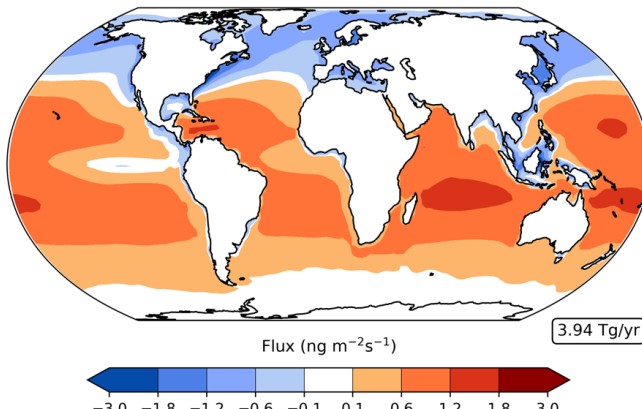

**Figure 7**. Same as Figure 4, for the ocean bidirectional fluxes.

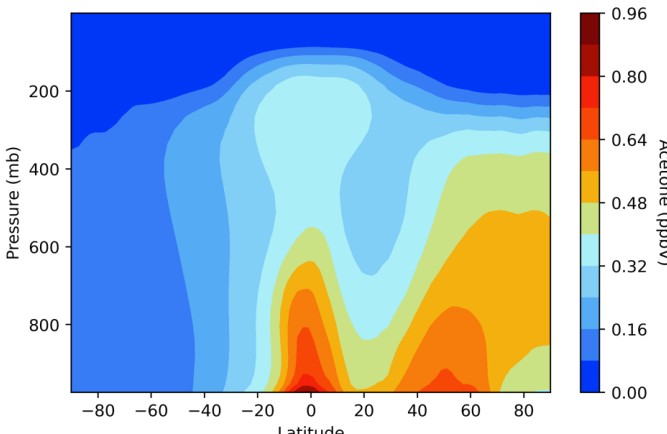

**Figure 8.** GISS ModelE2.1 vertical distribution of acetone air mixing ratios across latitudes in the Baseline simulation.





640

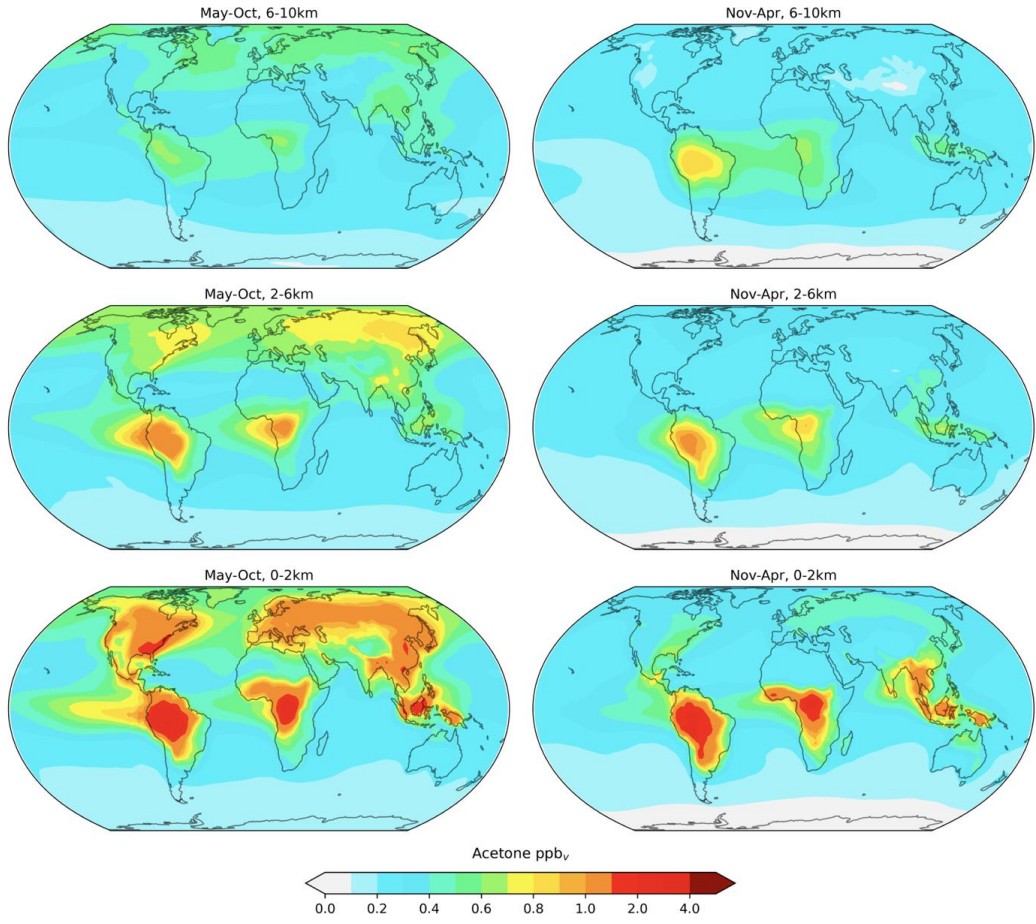

641

**Figure 9.** Baseline simulation acetone mixing ratios in the atmosphere at 0-2 km (bottom), 2-6 km (middle), and 6-10 km (top) for

the months of May-October (left) and November-April (right). The mixing ratios in the vertical were averaged with an arithmetic

mean. The choice of the slices and colors match those in Figure 1 by (Fischer et al., 2012).

645





**Figure 10.** Comparison between the GISS ModelE2.1 simulations (Baseline in purple and Nudged_ATom in blue) and the ATom-2 field measurements (January-February 2017). Individual data points are shown with grey dots, and their average values are shown in black, with error bars representing the one-sigma range of the averages.

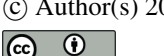

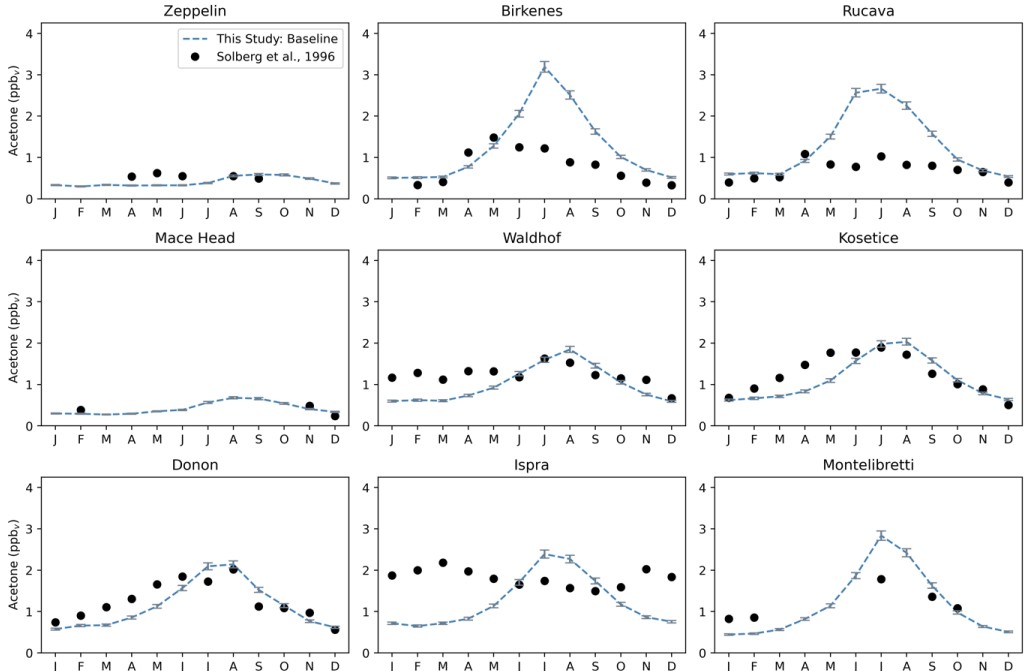

**Figure 11**. Acetone over twelve months at nine European sites, similar to that of Jacob et al. (2002). The modelled estimates of acetone at the surface from the Baseline simulation are shown in blue dashed lines and the grey error bars represent the one-sigma range of the modelled concentrations in the climatological mean of 5 years. Field measurements from Solberg et al., (1996) are shown as solid dots. Root mean squared error between the Baseline simulation and field measurements are (left to right, top to bottom): 0.1968, 0.8714, 0.8724, 0.0914, 0.3907, 0.3430, 0.3160, 0.9454, 0.5454.





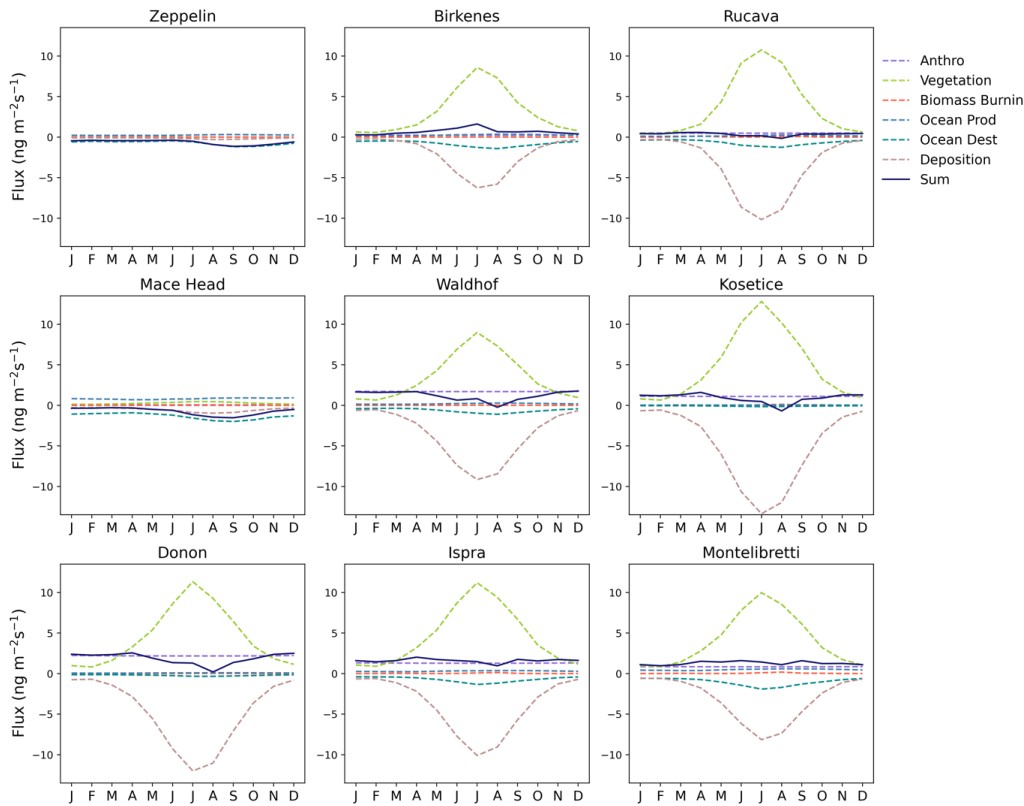

**Figure 12.** Contribution of acetone sources and sinks in the Baseline simulation over twelve months on the regional level (10˚ x 12.5˚ grid boxes) at nine European sites.





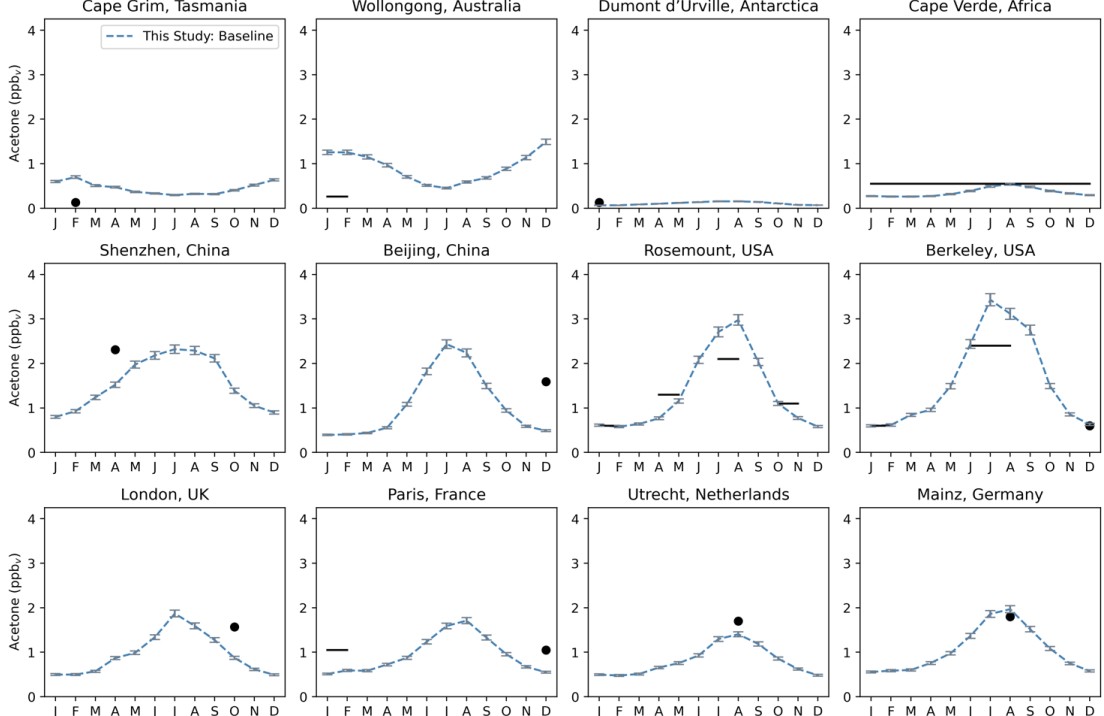

**Figure 13**. A similar plot to Figure 11 for various sites that do not have enough measurements to resolve seasonality (Australia, Antarctica, Africa, Asia, Europe, North America). The modelled estimates are overlaid with monthly (solid circles) or seasonal (solid lines) field measurements, as found in the literature.





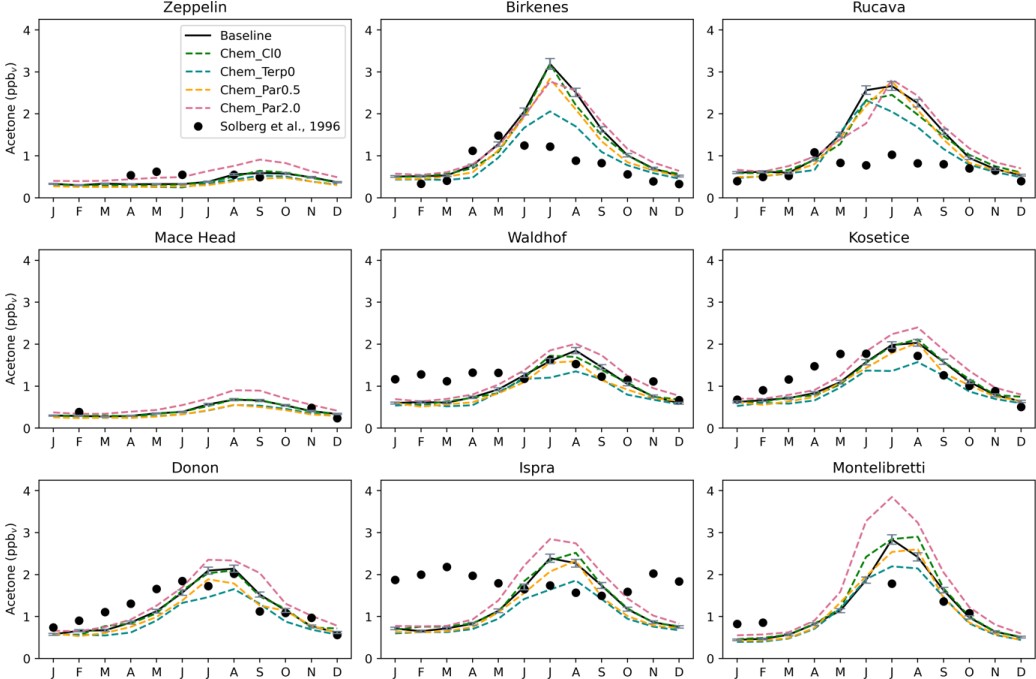

**Figure 14**. Same as Figure 11 with the chemistry sensitivity studies added. The sensitivity studies include removing the acetone + chlorine reaction (green line), removing the production of acetone from terpenes (blue line), halving the yield of acetone from paraffin (orange line), and doubling the yield of acetone from paraffin (pink line).



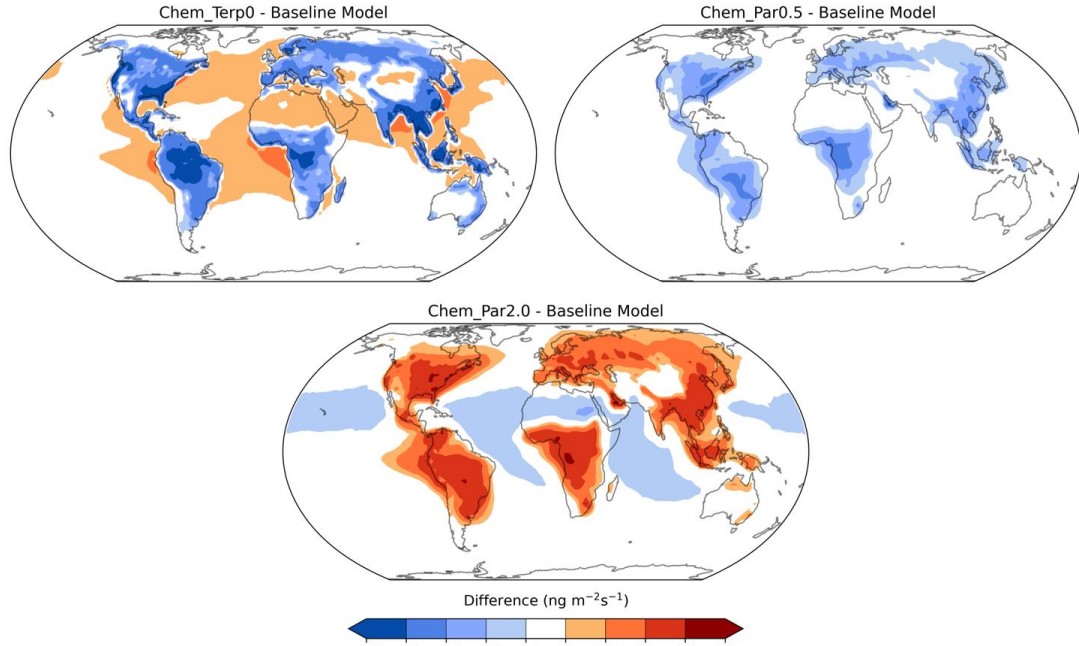

**Figure 15.** Chemistry sensitivities anomalies from Baseline, with red indicating an increase and blue indicating a decrease of the column-integrated net acetone chemistry flux. Nonlinear colorbars are used to better differentiate the details in the map. The fourth chemistry sensitivity study, Chem_Cl0, is omitted, since the changes everywhere are very small, less than 0.4 ng m$^{-2}$ s$^{-1}$.





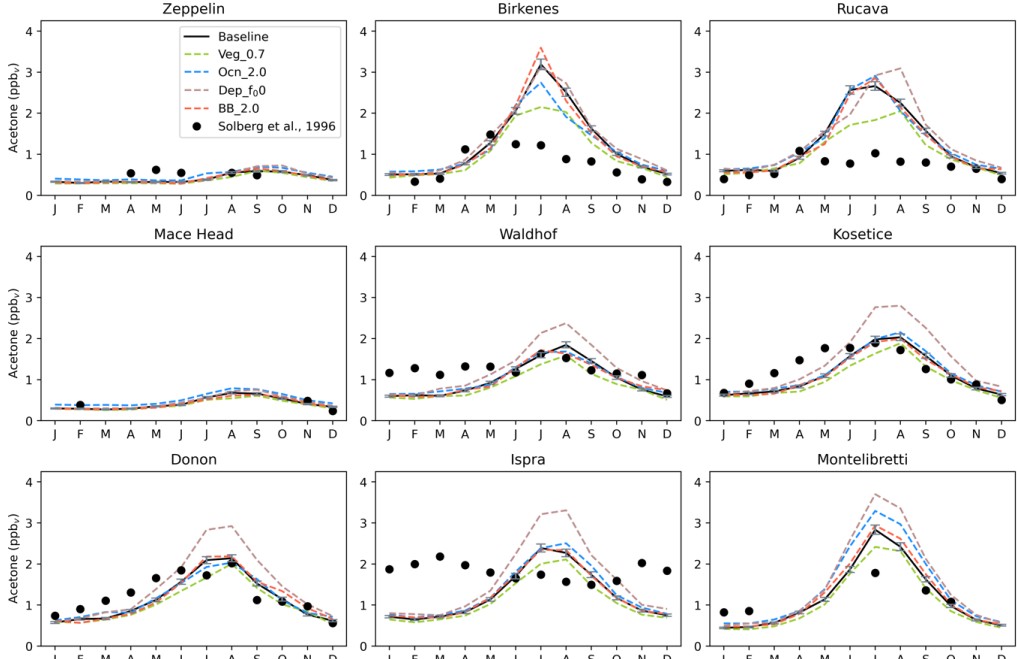

**Figure 16**. Same as Figure 11 with the terrestrial and oceanic sensitivity studies added. The sensitivity studies include reducing vegetation emissions to 0.7 acetone from MEGAN (light green line), doubling ocean acetone concentration (blue line), changing the reactivity factor for dry deposition (brown line), and doubling biomass burning emissions (orange line).

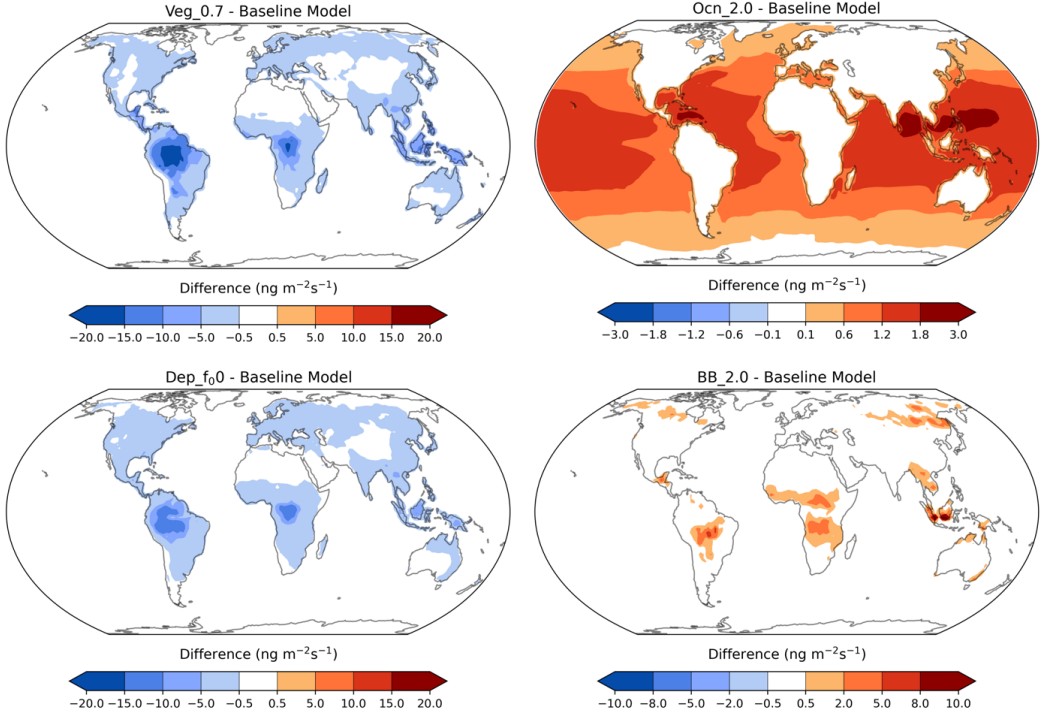

686

**Figure 17.** Acetone anomalies from the Baseline simulation for the vegetation (top left), ocean (top right), dry deposition (bottom

687    left) and biomass burning (bottom right) sensitivities, with red indicating an increase and blue indicating a decrease of the specific

689    flux. Nonlinear colorbars are used to better differentiate the details in the map.





**Figure 18**. A comparison between the GISS ModelE2.1 sensitivity simulations and the ATom-1 aircraft measurements (July-August 2016). Note that all sensitivities are to be compared against the Baseline simulation, not the Nudged_ATom one, but as shown earlier this makes very little difference in the comparison with observations (Figure 10).