# Peer review of "Assessing acetone for the GISS ModelE2.1 Earth system model"

_EGUsphere, 2023_

## Author Comment (AC1)

Dear Reviewer,

Thank you very much for your insightful comments. Below we have taken these comments (**in dark gray**) and have detailed our responses (**in blue**) and the amended manuscript text (***in blue italics***) if appropriate.

This paper presents an improved representation of acetone in GISS ModelE2.1 and evaluates it with observations. I don't understand why this is submitted to GMD considering that there is no originality or particular difficulty on the model development front – the approach replicates what has routinely been used by other models to describe acetone sources and sinks, without any tangible improvement. From a model development standpoint the implementation is trivial. I could see the value of documenting in GMD a major update to GISS ModelE chemistry affecting the simulation of species relevant to chemistry-climate interactions, but acetone is not important enough to rise to that standard, and the paper does not discuss how improved simulation of acetone affects the model's chemistry – presumably not much.

We understand your concern, however we believe our paper falls under the scope of the GMD in the following two ways, quoted from the journal's aims and scope page (https://www.geoscientific-model-development.net/about/aims_and_scope.html):

(1) The GMD considers manuscript types that are "geoscientific model descriptions, from statistical models to box models to GCMs"
(2) The GMD also considers manuscript types that are "development and technical papers, describing developments such as new parameterizations or technical aspects of running models such as the reproducibility of results"

Our paper discusses the evolution of acetone as a tracer in the NASA GISS ModelE2.1 GCM. We believe the paper constitutes a "geoscientific model description" as we describe how our model now simulates acetone sources and sinks and acetone chemistry in its atmosphere. We further believe that our paper qualifies as a development paper that describes "new parameterizations," as our updates from an outdated acetone scheme to one more in line with what is "routinely … used by other models" will have implications for the widely-used NASA GISS GCM.
The development included a lot of different components, so in our view it is not trivial: we had to implement a new photolysis calculation, bidirectional fluxes from the ocean, and alter our prescribed anthropogenic emissions to take into account the contribution of acetone. We also had to make decisions on the chemical destruction reactions to use, including its products. The sensitivity simulations we present in the manuscript clearly show that some decisions were not simple, and by presenting them we hope to help future researchers that will face the same questions in their models.

In my opinion, this paper is not a significant contribution to model development and is well below the standard of papers published in GMD. Publishing it might actually do some harm because it might be perceived as a new take on the global budget of acetone, which has not been revisited for some years, but in fact there is no innovation here aside from the evaluation with ATom measurements, and anthropogenic emission inventories are for 2000 which is dated. The evaluation with ATom is in my view the most interesting part of the paper but the interpretation is cursory.

Our intention is not to develop a new take on the global budget of acetone, but rather document our improvements on the representation of acetone in the NASA GISS GCM, which would otherwise not be readily available to the scientific community, and to refer to modeling and field studies to validate

our improvements. To make our intentions more clear in the manuscript, Section 2 has been updated to include the following (see line 101):

*Here we implement acetone in the GISS ModelE2.1 based on the literature rather than developing a new parameterization.*

We agree that the evaluation with ATom measurements is promising, and for this reason we have expanded our discussion on it (see response to specific comment #7 below).

A few specific comments:
1. Line 39: acetone is not highly water-soluble by atmospheric standards.

Thank you for pointing this out. We have adjusted lines 38-39 to remove the word "high":

*Wet deposition occurs within and below clouds due to the solubility of acetone, and depends on its Henry's Law coefficient (Benkelberg et al., 1995).*

2. Line 40: oxidation of acetone by OH is not a net source of radicals.

Thank you for pointing this out. We have adjusted line 40 to only mention photolysis:

*Chemical loss of acetone forms radicals through photolysis.*

3. Line 170: what spectroscopic and quantum yield data are used for acetone photolysis? There has been some work done on that recently.

To clarify the spectroscopic and quantum yield data we used for acetone photolysis, we have added the following in Section 2.3.2 (see lines 208-211):

*The spectroscopic data used for acetone photolysis is from JPL 2010 (Sander et al., 2011) and mapped onto Fast-J version 6.8d's wavelength intervals (Neu et al., 2007). The quantum yields are pressure and temperature dependent and thus vary with altitude and location. For example, in a standard atmosphere the ratio of the yield of CO to $CH_3CO$ decreases from 0.28 at the surface to 0.18 at 4 km altitude.*

4. Line 180: Where does the fixed ocean concentration of 15 nM come from? What is the justification for assuming a fixed concentration? This obviously effects the sign and geographical distribution of the air-sea acetone fluxes mentioned in the abstract.

We chose a constant ocean acetone concentration of 15 nM following the GEOS-CHEM model's implementation (Fischer et al., 2012). See page 2 paragraph 8 of Fischer et al., where the researchers observe a collection of data points of ocean acetone concentration and conclude that "The data do not show evident seasonal or spatial patterns that would warrant a more detailed treatment." We have updated our manuscript to mention this reference in Section 2.4 (see lines 216-220):

*The atmospheric source from ocean water and sink from the atmosphere are calculated assuming a constant concentration of acetone in water (of 15 nM), the lower boundary layer atmospheric concentration, and the total transfer velocity (a combination of water-side and air-side transfer velocities). The constant concentration of 15 nM follows the implementation by Fischer et al. (2012) in*

*the GEOS-CHEM model, who looked at observations and did not find a strong reasoning to make the concentration vary seasonally or spatially.*

Furthermore, we would like to mention that the ocean acetone concentration of our model is something we are also concerned about, as we have a sensitivity run that tests doubling this concentration to 30 nM. In future work, we may replace the constant 15 nM with prognostic water concentration, but this goes beyond the scope of this paper.

5. Lines 228-229, 257-258: documenting the improvement over the previous GISS ModelE parameterization of acetone is of little interest considering that the previous parameterization was so crude by current model standards.

Referring back to the scope of the GMD (see response above), we do believe that the fact that the previous parameterization was crude is a valid justification to document the improvement. The previous acetone scheme was part of a GISS model that was already being widely used both in intercomparison studies (CMIP, HTAP, AeroCom, to name a few) and by the community, since it is a public model used even outside GISS, so these improvements are not insignificant or of no interest to be documented.

We kept the statistics comparing the acetone concentrations at the surface (see lines 286-288) because we believe these are significant. As per your suggestion, we did agree to take out Figure S3 from the supplement, as the plot was trivial and did not add much to our comparison of the prior and updated acetone schemes. The remaining figures in the supplement were renamed and their citations were updated in the manuscript.

6. Line 325: why would there be non-linearities in the system? I presume that the acetone simulations use the full chemistry mechanism (although that's not clear – it could also be done with archived OH fields and production rates) but since acetone is in general a minor player in oxidant chemistry I don't see why there would be significant non-linearities. The asymmetry in response to doubling/halving is not necessarily a sign of non-linearity in the presence of other sources/sinks. If there is indeed significant non-linearity the authors should explain why.

Yes, we are using the full chemistry and not the archived OH fields. We have clarified this in the manuscript by adding the following in the general model description in Section 2 (see lines 104-105):

*Acetone simulations use full chemistry and not archived OH fields.*

In response to the nonlinearity concerns, the chemistry is nonlinear, and the fact that no major nonlinearities appear in our results does not mean they do not exist. They are simply not triggered, because acetone is not a central species in the chemistry of the atmosphere. We would also like to mention that we simply stated that we noticed a nonlinear response (that our sensitivity study confirmed that yield from paraffin was not a linear control); we did not say this was a "significant" nonlinearity in the system.

7. Lines 383-384: analysis of the evaluation with ATom observation is limited to uninstructive throwaway statements. The authors should do better. Would correlations with other chemical species be insightful?

We have expanded the ATom analysis presented in sections 3.3 and 3.5.3 and the figures in several ways. We have added the root mean square error (RMSE) of each simulation performed for all regions present in Figures 10, 18, S3-S5, and S15-S17. We have also augmented the discussion in sections 3.3

and 3.5.3 as described below, to make it more quantitative, and also present the results in a more systematic and informative way. We decided to not compare with other chemical species, since this would need to include a full analysis of the gas-phase chemistry results of the model to be complete, which is beyond the scope of the manuscript. …

[revised manuscript text omitted]

---

## Author Comment (AC2)

Dear Reviewer,

Thank you very much for your insightful comments. Below we have taken these comments (**in dark gray**) and have detailed our responses (**in blue**) and the amended manuscript text (***in blue italics***) if appropriate.

Acetone is an important molecule in troposphere chemistry cycles and to model it correctly a rather sophisticated scheme is needed because of the complex sources and sinks including biological, chemical and physical processes. Also to note is that acetone has been well studied but recent widespread measurements provide an opportunity to further refine models.

Significant effort was put into the work described in this paper and the authors made good progress toward improving their GISS model representation of acetone. If this is sufficient for publication in this journal then fine and I can provide a more thorough review. However if, to be published in this journal, (which I am not very familiar with) there is a requirement that the paper provides an important contribution to our understanding of acetone in the atmosphere then it falls short because, although significant progress was made on this model, the work does not represent an improvement over previous modeling efforts. If this is the case I would recommend rejecting the paper. The editor will have to make the call on this.

We understand your concern, however we believe our paper falls under the scope of the GMD in the following two ways, quoted from the journal's aims and scope page (https://www.geoscientific-model-development.net/about/aims_and_scope.html):

(1) The GMD considers manuscript types that are "geoscientific model descriptions, from statistical models to box models to GCMs"
(2) The GMD also considers manuscript types that are "development and technical papers, describing developments such as new parameterizations or technical aspects of running models such as the reproducibility of results"

Our paper discusses the evolution of acetone as a tracer in the NASA GISS ModelE2.1 GCM. We believe the paper constitutes a "geoscientific model description" as we describe how our model now simulates acetone sources and sinks and acetone chemistry in its atmosphere. We further believe that our paper qualifies as a development paper that describes "new parameterizations," as our updates from an outdated acetone scheme to one more in line with what is "routinely … used by other models" will have implications for the widely-used NASA GISS GCM.
The development included a lot of different components, so in our view it is not trivial: we had to implement a new photolysis calculation, bidirectional fluxes from the ocean, and alter our prescribed anthropogenic emissions to take into account the contribution of acetone. We also had to make decisions on the chemical destruction reactions to use, including its products. The sensitivity simulations we present in the manuscript clearly show that some decisions were not simple, and by presenting them we hope to help future researchers that will face the same questions in their models.

There are some things in the paper that cause concern on my part. One is that the derived chemical lifetime from the expression (burden (Tg) /sink (Tg/year)) is too short. (Also, the burden units are wrong on line 203). The kinetics of the reaction of acetone with OH is well known and the authors have the correct expressions on line 169. Based on the OH reaction, if one assumes that the diurnal [OH] is 0.7E * 05, then the chemical lifetime derived due to OH reaction is 95 days. The lifetime with respect to Cl is significantly longer than this and the lifetime with respect to photolysis is also longer

than this. So combined the actual atmospheric chemical lifetime is most likely > 40 days. However the derived value in the paper from burden/sink is 20 days which to me indicates a problem with the formulation of either the burden or the sink terms that needs to be investigated further.

Thank you for pointing out the incorrect burden units on line 203. The line (now 255) is now fixed to say:

*The GISS ModelE2.1 Baseline simulation estimates the burden to be 2.93 Tg.*

We are unsure why the reviewer assumed an OH concentration of 7.E4 molecules cm-3. From our Baseline simulation we find a 5 year average and area-weighted spatial average of about 8.3E5 molecules cm-3 near the surface (and a bit more than that moving upwards in the troposphere), so more than an order of magnitude difference vs. that assumed value. This OH would have a strong influence on the methane lifetime in our model, which we find to be about 9.6 years using the same metric as we used to calculate acetone's lifetime. This is in good agreement with the IPCC AR6 report (Intergovernmental Panel On Climate Change, 2023) page 701, which estimates a methane lifetime constrained by observations of 9.1 ± 0.9 years (all loss mechanisms, but they estimate tropospheric OH as 90% of the sink.)

We have tried to explicitly place our acetone lifetime (along with all sources and sinks) in the context of the literature in Table 2 and Figure 2. This indeed shows that our model implementation leads to a short chemical lifetime of acetone compared to other studies (which we acknowledge in the text as well in lines 269-271), but not nearly as short as the reviewer's estimates above would imply. We are glad the reviewer mentioned this, as it allowed us to notice an accidental deletion of the values found in this study in Table 2, which may have caused some confusion and has now been rectified.

Another shortcoming is the chemical representation in the model. The chemical scheme appears to be rudimentary and also seems to have errors.

The authors state: "Initial tests using a yield of 0.72 resulted in an overestimated chemistry source, leading us to re-evaluate this yield for the specific mixture of VOCs represented in the GISS ModelE2.1. Estimated mole fractions of propane (11%), butane (22%) and pentane (21%) in anthropogenic emissions were multiplied by each compound's acetone molar yield (0.73, 0.95, 0.63, respectively), determining that 42% of paraffin from anthropogenic sources becomes acetone".

The authors don't reference where they got their estimated mole fractions from anthropogenic emissions or where they got the molar yield of acetone for each compound. Furthermore the molar yields are incorrect. Although propane does in fact form acetone with a significant yield, butane and pentane do not.

Thank you for bringing this to our attention – we have updated Section 2.3.1 to better reference where we obtained our mole fractions. Our updated section in the manuscript reads as follows (see lines 168-177):

*Our model's anthropogenic emissions of paraffin is based on an aggregation of selected VOC groups. Based on year 2019 emissions of the O'Rourke et al. (2021) dataset, we emit paraffin that is about 11% propane by mole, 22% butane and 21% pentane. Multiplying these by each VOC's acetone molar yield (0.73, 0.95, 0.63, respectively), we estimate that 42% of paraffin from anthropogenic sources becomes acetone in our model. Paraffin biomass burning emissions, estimated from year 2020 of SSP3_70*

*emissions (Riahi et al., 2017; Fujimori et al., 2017) contain mole fractions for propane of 9% and higher alkanes of 23%, and when multiplied by acetone molar yields of 0.73 and 0.79, respectively, suggest that about 25% of paraffin from biomass burning sources becomes acetone in our model. The molar yields used in these calculations were derived with suggestions from the literature (Fischbeck et al., 2017; Jacob et al., 2002; Weimer et al., 2017). Refer to the manuscript supplement for a more detailed breakdown. Overall, an average of the 42% anthropogenic paraffin and 25% biomass burning paraffin was used to conclude that approximately 35% of paraffin from emissions becomes acetone, leading to our refinement of the molar yield in Eq. (1) to 0.35.*

The more detailed breakdown in the manuscript supplement reads as follows (lines 10-16):

*The acetone molar yields of propane, butane, pentane, and higher alkanes were derived with suggestions from the literature (Fischbeck et al., 2017; Jacob et al., 2002; Weimer et al., 2017). We used a molar yield of 0.73 for propane, derived by averaging 0.72 from Jacob et al. (2002) and 0.736 from Weimer et al. (2017). Our molar yield of 0.95 for butane was derived by averaging 0.96 from Fischbeck et al. (2017) and 0.93 from Jacob et al. (2002). Our molar yield of 0.63 for pentane was derived by averaging 0.72 from Fischbeck et al. (2017) and 0.53 from Jacob et al. (2002). Finally, we used a molar yield of 0.79 for higher alkanes, derived from averaging the following four values: 0.96 for isobutane and 0.72 for isopentane in Fischbeck et al. (2017), and 0.93 for isobutane and 0.53 for isopentane in Jacob et al. (2002).*

In summary, I refer to my comments above in the second paragraph.

**References**

Fischer, E. V., Jacob, D. J., Millet, D. B., Yantosca, R. M., and Mao, J.: The role of the ocean in the global atmospheric budget of acetone, Geophys. Res. Lett., 39, https://doi.org/10.1029/2011GL050086, 2012.

Fujimori, S., Hasegawa, T., Masui, T., Takahashi, K., Herran, D. S., Dai, H., Hijioka, Y., and Kainuma, M.: SSP3: AIM implementation of Shared Socioeconomic Pathways, Glob. Environ. Change, 42, 268–283, https://doi.org/10.1016/j.gloenvcha.2016.06.009, 2017.

Intergovernmental Panel On Climate Change: Climate Change 2021 – The Physical Science Basis: Working Group I Contribution to the Sixth Assessment Report of the Intergovernmental Panel on Climate Change, 1st ed., Cambridge University Press, https://doi.org/10.1017/9781009157896, 2023.

O'Rourke, P. R., Smith, S. J., Mott, A., Ahsan, H., McDuffie, E. E., Crippa, M., Klimont, Z., McDonald, B., Wang, S., Nicholson, M. B., Feng, L., and Hoesly, R. M.: CEDS v_2021_04_21 Release Emission Data, https://doi.org/10.5281/zenodo.4741285, 2021.

Riahi, K., van Vuuren, D. P., Kriegler, E., Edmonds, J., O'Neill, B. C., Fujimori, S., Bauer, N., Calvin, K., Dellink, R., Fricko, O., Lutz, W., Popp, A., Cuaresma, J. C., Kc, S., Leimbach, M., Jiang, L., Kram,

T., Rao, S., Emmerling, J., Ebi, K., Hasegawa, T., Havlik, P., Humpenöder, F., Da Silva, L. A., Smith, S., Stehfest, E., Bosetti, V., Eom, J., Gernaat, D., Masui, T., Rogelj, J., Strefler, J., Drouet, L., Krey, V., Luderer, G., Harmsen, M., Takahashi, K., Baumstark, L., Doelman, J. C., Kainuma, M., Klimont, Z., Marangoni, G., Lotze-Campen, H., Obersteiner, M., Tabeau, A., and Tavoni, M.: The Shared Socioeconomic Pathways and their energy, land use, and greenhouse gas emissions implications: An overview, Glob. Environ. Change, 42, 153–168, https://doi.org/10.1016/j.gloenvcha.2016.05.009, 2017.

---

## Editor Decision (ED1)

[revised manuscript text omitted]

0.72 from Jacob et al. (2002) and 0.736 from Weimer et al. (2017). Our molar yield of 0.95 for butane was derived by averaging

0.96 from Fischbeck et al. (2017) and 0.93 from Jacob et al. (2002). Our molar yield of 0.63 for pentane was derived by averaging 0.72 from Fischbeck et al. (2017) and 0.53 from Jacob et al. (2002). Finally, we used a molar yield of 0.79 for higher alkanes, derived from averaging the following four values: 0.96 for isobutane and 0.72 for isopentane in Fischbeck et al. (2017), and 0.93 for isobutane and 0.53 for isopentane in Jacob et al. (2002).

**17    3.1 Global acetone budget and burden**

[Figure]

**Figure S1.** The data recorded in the literature was used to determine a mean for each budget flux value, and the distance from that mean for each paper was expressed as a z-score (Arnold et al., 2005; Beale et al., 2013; Brewer et al., 2017; Dufour et al.,

2016; Elias et al., 2011; Fischer et al., 2012; Folberth et al., 2006; Guenther et al., 2012; Jacob et al., 2002; Khan et al., 2015;

Marandino et al., 2005; Singh et al., 2000, 2004; Wang et al., 2020).The z-scores for the literature are in shown as light-blue bars, and the Baseline model's z-score is highlighted in yellow.

 **3.2 Spatial distribution of acetone**

[Figure]

**Figure S2.** Net oceanic acetone fluxes in the Baseline simulation for December-February (top left), March-May (top right), June-August (bottom left), and September-November (bottom right), with red indicating a net source and blue indicating a net sink. Nonlinear colorbars are used to better differentiate the details in the map. The weighted global means of the net ocean fluxes are shown in boxes on the lower right.

[Figure]

**Figure S3.** Comparison between the GISS ModelE2.1 simulations (Baseline in purple and Nudged_ATom in blue) and the ATom-1 field measurements (July-August 2016). Individual data points are shown with grey dots, and their average values are shown in black, with error bars representing the one-sigma range of the averages. The root mean square error (RMSE) of each simulation is shown at the top right of each plot.

[Figure]

**Figure S4.** Similar to Figure S3, except for the ATom-3 field measurements (September-October 2017).

[Figure]

**Figure S5.** Similar to Figure S3, except for the ATom-4 field measurements (April-May 2018).

**3.4 Seasonality of acetone**

[Figure]

**Figure S6.** GISS ModelE2.1 spatial distribution of annual mean acetone at surface for the Baseline simulation in Europe over twelve months. Filled circles represent data from field measurements from Solberg et al. (1996). A nonlinear colorbar is used to better differentiate the details in the map.

**3.5 Sensitivity Studies**

[Figure]

**Figure S7.** Total atmospheric burden, fluxes, and lifetimes of acetone from the literature (shown in boxes and whiskers with outliers as open circles) (Arnold et al., 2005; Beale et al., 2013; Brewer et al., 2017; Dufour et al., 2016; Elias et al., 2011;

Fischer et al., 2012; Folberth et al., 2006; Guenther et al., 2012; Jacob et al., 2002; Khan et al., 2015; Marandino et al., 2005;

Singh et al., 2000, 2004; Wang et al., 2020), values from GISS ModelE2.1 Baseline simulation (solid blue circles), and values from the Chem_Terp0 sensitivity study (green circles).

[Figure]

**Figure S8**. Similar to Figure S7, except values from the Chem_Par0.5 sensitivity study as green circles.

[Figure]

**Figure S9.** Similar to Figure S7, except values from the Chem_Par2.0 sensitivity study as green circles.

[Figure]

**Figure S10.** Similar to Figure S7, except values from the Veg_0.7 sensitivity study as green circles.

[Figure]

**Figure S11.** Similar to Figure S7, except values from the Ocn_2.0 sensitivity study as green circles.

[Figure]

**Figure S12.** Similar to Figure S7, except values from the Dep_$f_0$0 sensitivity study as green circles.

**3.5.1 Chemistry**

[Figure]

**Figure S13.** Acetone over twelve months for various sites that do not have enough measurements to resolve seasonality (Australia, Antarctica, Africa, Asia, Europe, North America). The modelled estimates of acetone at the surface from the Baseline simulation are shown as solid black lines, and the sensitivity studies are as follows: removing the acetone + chlorine reaction (dashed green lines), removing the production of acetone from terpenes (dashed blue lines), halving the yield of acetone from paraffin (dashed orange lines), and doubling the yield of acetone from paraffin (dashed pink lines). The modelled estimates are overlaid with monthly (solid circles) or seasonal (solid lines) field measurements, as found in the literature (de Gouw et al.,

2004; Dolgorouky et al., 2012; Galbally et al., 2007; Guérette et al., 2019; Hu et al., 2013; Huang et al., 2020; Langford et al.,

2010; Legrand et al., 2012; Li et al., 2019; Read et al., 2012; Schade and Goldstein, 2006).

**3.5.2 Terrestrial and oceanic fluxes**

[Figure]

**Figure S14.** Similar to Figure S13, but with the terrestrial and oceanic sensitivity studies added. The modelled estimates of acetone at the surface from the Baseline simulation are shown as solid black lines, and the sensitivity studies are as follows:

reducing vegetation emissions to 0.7 acetone from MEGAN (dashed light-green line), doubling ocean acetone concentration (dashed blue line), changing the reactivity factor for dry deposition (dashed brown line), and doubling biomass burning emissions (dashed orange line). Field measurements from Solberg et al., (1996) are shown as solid black dots. The modelled estimates are overlaid with monthly (solid circles) or seasonal (solid lines) field measurements, as found in the literature (de

Gouw et al., 2004; Dolgorouky et al., 2012; Galbally et al., 2007; Guérette et al., 2019; Hu et al., 2013; Huang et al., 2020;

Langford et al., 2010; Legrand et al., 2012; Li et al., 2019; Read et al., 2012; Schade and Goldstein, 2006).

**3.5.3 ATom comparisons**

[Figure]

**Figure S15.** Similar to Figure S3, except a comparison between the GISS ModelE2.1 sensitivity simulations and the ATom-2 aircraft measurements (January-February 2017). Individual data points are shown with grey dots, and their average values are shown in black, with error bars representing the one-sigma range of the averages. The root mean square error (RMSE) of each simulation is shown at the top right of each plot. Note that all sensitivities are to be compared against the Baseline simulation.

[Figure]

**Figure S16.** Similar to Figure S15, except for the ATom-3 field measurements (September-October 2017).

[Figure]

**Figure S17.** Similar to Figure S15, except for the ATom-4 field measurements (April-May 2018).

**References**

[revised manuscript text omitted]

---

## Author Response (AR2)

We would like to thank the editor for their insightful comments. Below we have taken these comments (**in dark gray**) and have detailed our responses (**in blue**) and the amended manuscript text (*in blue italics*) if appropriate. Please note that the line numbers referenced below correspond to the manuscript document *with* tracked changes visible.
* * *
**Public justification (visible to the public if the article is accepted and published):**
Dear Kostas Tsigaridis and co-authors,

Thank you for submitting the revised version of your manuscript. I find the reviewers' comments have resulted in a perceptible increase in the manuscript quality. I do not share reviewers' criticism regarding the suitability of this study publication for GMD (we need to document model development in either way), however I very second them in the opinion that the very "climatological" design of the model simulations and limited analysis depth do not allow any new insights on atmospheric acetone. The latter is also the reason why I cannot suggest you resubmitting this manuscript to, e.g., the ACP. Furthermore, the presented analysis is overburden with figures and statements, yet it lacks in-depth discussion on processes (e.g. chemical feedbacks claimed) and thus offers only limited insights about what could be improved in the future in modelling acetone with this or another modelling system. However, I see the way for this study being published here, provided that you address all the specific comments outlined below (please find copies of the manuscript/supplement with respective statements highlighted). These address presentation issues, overstated claims and reduction of the amount of presented material. Provided that all these remarks are addressed, I will be ready to reconsider the publication of the revised manuscript.

With best regards,
S. Gromov

General remarks
L23-24 This is an overstatement, as you do not present an analysis offering a "crucial step to umderstding the role of acetone in the atmosphere". Please reformulate accordingly, i.e. stating that the new implementation results in acetone budget/turnover in line with previous studies.
We have reformulated this last sentence of our abstract as follows (lines 23-24):
*Overall, our implementation is one that corroborates with previous studies and marks a significant improvement to the development of the acetone tracer in the GISS ModelE2.1.*

L533-534 Neither reviewers nor myself can confirm your being extensive, please reformulate accordingly or remove the sentence.
The word "extensive" has been removed from the sentence (line 939).

L540-541 Same as above, consider removing this sentence.
This sentence has been removed.

Specific remarks
Main text

L98-99  Please add which data product the ocean surface conditions are derived from.
The last sentence of the Section 2 now reads as follows (lines 173-174):
*This simulation employed nudged winds from MERRA2 (Gelaro et al., 2017), ocean surface conditions from PCMDI-AMIP 1.1.4 for 2016-2017 (Taylor et al., 2000) and from Hadley Center HadISST1.1 for*

*2018 (Met Office, Hadley Centre, 2006), and trace gas and aerosol emissions changing with time during 2016-2018.*

L124,130,231    Please use correct statistical term, i.e. you are presenting interannual variations, not the variability (which is the measure of the true population spread you do not have information on).
The word "variability" has been replaced with "variation" (lines 199, 205, 389).

L126    Same as above, perhaps use "differences between models".
The word "variability" has been replaced with "disagreements" (line 201).

L131    From the figure caption it is ambiguous whether C3H6 or acetone emissions are shown. Do you present fluxes or acetone calculated from C3H6 fluxes? With which yield?
Apologies for the typo in the figure caption. The sentence has been fixed to say "NMVOC-C3H6O" (line 205).

L152,167,etc.    Please use standard mathematical notation (base 10 instead of engineering E+-exponent, e instead of exp()) in reaction rate expressions throughout the manuscript.
Our reaction rate expressions have been rewritten in standard mathematical notation (lines 248, 262, 263).

L182-182    To be precise, all atmospheric reactions are pressure- and temperature-dependent simply due to changes in air density. Please use "reaction rate" instead of just "reaction".
This section has been re-written as follows (lines 266-270):
*The spectroscopic data used for acetone photolysis is from JPL 2010 (Sander et al., 2011) and mapped onto Fast-J version 6.8d's wavelength intervals (Neu et al., 2007). The photolysis cross section for Eq. 5 is pressure-dependent while that of Eq. 6 is temperature-dependent, leading to variation in yields with altitude and location. For example, in a standard atmosphere the ratio of the yield of CO to CH3CO decreases from 0.28 at the surface to 0.18 at 4 km altitude.*

L183    You imply photolysis quantum yields?
This section has been rewritten to instead refer to cross sections. See the response above, lines 266-270.

L198-199    Please refomulate sentence clearer. You are interested in sensitivity of simulated acetone burden/turnover to perturbations in a given parameter, I assume.
The sentence has been re-written as follows (lines 283-284).
*Specifically, we were interested in seeing the sensitivity of simulated acetone to artificial perturbations in given parameters.*

L199    By "sources" you rather mean "production", please avoid potential misunderstanding as "surface sources" or "direct sources" (e.g. L202).
The word "sources" was replaced with the word "production" (line 284).

L218-220    Please do not use negative figures for categories already implying removal (chem. sink, deposition) to avoid ambiguity. Use brackets in negative net fluxes ranges, e.g. −(a−b).
Table 2 has been modified so that the "Global Deposition," "Ocean Sink" and "Chem Sink" rows do not use negative figures. The bracket notation is now used for the negative flux range in the Net Chemistry row (lines 319-378).

L237-238   Please reformulate the sentence. It is not clear what you want to state. As of now, it appears that this comparison is not intended to corroborate your estimates, then what is it presented for?
We have removed this sentence.

L265   Omit either "negative" or the minus sign whilte reporting the flux here.
The word "negative" has been removed from this sentence (line 430).

L262-270   Please remove Figure 4 (it contains the summary of the features seen in Figure 5 and described in this statement. Please reformulate the paragraph accordingly.
Figure 4 has been removed and the paragraphs and all subsequent figure labels have been adjusted accordingly.

L291-300   Please combine Figures 6 and 7 into one, i.e. showing loss, source and net in one row next to each other.
Figures 6 and 7 (now Figure 5) have been merged, and all subsequent figure labels have been adjusted accordingly.

L305-306   How high is "as high"? You likely imply "troposphere", not "atmosphere"?
The word "atmosphere" was replaced with the word "troposphere" (line 488).

L317-318   It is an erroneous approach to average mixing ratios simply arithmetically because the air density changes significantly in the regarder altitude ranges. Please recalculate with mass-weighting of mixing ratios and replot.
We have redone this plot so that it is calculated using a mass-weighting of acetone mixing ratios instead of the arithmetic mean. With the exception of slight changes in the contours in 0-2 km subplots, the figure has not qualitatively changed. Therefore we did not find the need to edit the main text referring to this figure. The figure caption has been rewritten to reflect these changes (lines 538-541):
*Figure 7. Baseline simulation acetone mixing ratios in the atmosphere at approximately 0-2 km (bottom), 2-6 km (middle), and 6-10 km (top) for the months of May-October (left) and November-April (right). The average mixing ratios over these broad altitude layers are weighted by the air mass in the model layers they contain. The choice of the slices and colors match those in Figure 1 by (Fischer et al., 2012).*

L355-361   Please quote RMSE values in respective plot panels in the figure, similar to that in Figure 10.
The RMSE values of each respective subplot are now displayed on the figure (line 591).

L365   Please use the regular equal sign ("="), use of "approximately equal" is inappropriate here.
The approximately equal sign has been replaced with a regular equal sign (line 600).

L370-393   Please move Figures 12 and 13 to the Supplement.
Figures 12 and 13 are now Figures S7 and S8 in the 'Seasonality of Acetone' section of the supplement.

L422-432   The analysis of chemistry sensitivity studies offers no insights (read is very shallow), at least explicate the mechanism responsible for the "feedbacks" (L424-428). Please mention results for Chem_Cl0 in the paragraph text, not the Figure 15 caption. Please move Figure 15 to the supplement.

Figure 15 is now Figure S16 in the 'Chemistry' section of the supplement. The discussion of this plot in the manuscript has been adjusted to include the results for Chem_Cl0, and the analysis of feedbacks has been expanded (lines 711-752):

*The spatial distribution differences between the chemistry sensitivity studies and the Baseline simulation show some interesting patterns (Figure S16). Removing the production of acetone from terpenes oxidation in the Chem_Terp0 simulation decreased acetone over the continents, and especially over tropical and boreal forests which are where terpenes are emitted. This change also increased acetone concentrations over the oceans due to chemical composition changes downwind that result from the change of terpenes oxidation products (Figure S16, top left). Halving production of acetone from paraffin oxidation in the Chem_Par0.5 simulation only decreased acetone concentrations over the continents, while doubling it in the Chem_Par2.0 simulation increased acetone concentrations over the continents and strengthened acetone destruction over the tropical oceans (Figure S16, top right and bottom middle, respectively). Setting the acetone + chlorine reaction rate to 0 in the Chem_Cl0 simulation resulted in negligible changes across the globe (anomalies of $<0.4$ ng m$^{-2}$ s$^{-1}$).*

L433-444    Move Figure 16 to the Supplement to accompany Fig. S14.
Figure 16 is now Figure S17 in the 'Terrestrial and oceanic fluxes' section of the supplement.

L459-462    Reformulate the sentence – you iterate "increase" 11 times here. E.g. strongest increase is seen in Ispra (38.4%), Kosetice (…) etc.
We reformulated the sentence to remove all the redundant uses of the word "increase" (lines 774-775).

L503,510,517    Please use "observations" instead of "measurements" (the latter are the act of measuring, so you can't "underestimate" these).
All occurrences of the word "measurements" were replaced with the word "observations" (lines 902, 916, 923).

L521-523    This is a self-contradictory statement, please reformulate.
The sentence has been reformulated, and the incorrect statement "underestimated sink" has been fixed to say "overestimated sink" (line 928).

L527    Please amend the sentence, which "some conditions" are implied?
The vague term "conditions" was replaced with a more descriptive "sensitivity simulations" (line 933).

L557    Make sure the URL data to the Zenodo publication points at doi.org or zenodo.org (currently it points at some address starting with urldefense.com)
We updated our manuscript so that the URL is plain text and does not point to the address starting with urldefense.com (line 972).

Figures 10, 18, S3−S5, S15
        Currently the symbols for individual observations are barely distinguishable either on the screen or in hard-copy. Please re-plot figures with thicker symbols (you may also use pluses "+") in better visible colours (e.g. red). Please use vector graphics for these comparison plots.
All ATom plots have been re-done using darker gray and a "+" symbol for the individual observations. They were all generated using vector graphics.

Supplement
Please number the sections sequentially, e.g. S1, S2 etc
The numbers in the section headers were removed so that all figures are numbered sequentially.

**References**

Fischer, E. V., Jacob, D. J., Millet, D. B., Yantosca, R. M., and Mao, J.: The role of the ocean in the global atmospheric budget of acetone, Geophys. Res. Lett., 39, https://doi.org/10.1029/2011GL050086, 2012.

Met Office, Hadley Centre: HadISST 1.1 - Global sea-Ice coverage and SST (1870-Present), [Internet]. NCAS British Atmospheric Data Centre 2006, April 3, 2021. Available from http://badc.nerc.ac.uk/view/badc.nerc.ac.uk__ATOM__dataent_hadisst

Neu, J. L., Prather, M. J., and Penner, J. E.: Global atmospheric chemistry: Integrating over fractional cloud cover, J. Geophys. Res. Atmospheres, 112, 2006JD008007, https://doi.org/10.1029/2006JD008007, 2007.

Taylor, K., Williamson, D., and Zwiers, F.: The sea surface temperature and sea ice concentration boundary conditions for AMIP II simulations, PCMDI Report 60, Program for Climate Model Diagnosis and Intercomparison, Lawrence Livermore National Laboratory, 2000.

Sander, S. P., J Abbatt, J. R. Barker, J. B. Burkholder, R. R. Friedl, D. M. Golden, R. E. Huie, C. E. Kolb, M. J. Kurylo, G. K. Moortgat, V. L. Orkin, and P. H. Wine: Chemical Kinetics and Photochemical Data for Use in Atmospheric Studies Evaluation No. 17, JPL Publication 10-6, Jet Propulsion Laboratory, Pasadena, 2011.